# *Drosophila* larval to pupal switch under nutrient stress requires IP$_3$R/Ca$^{2+}$ signalling in glutamatergic interneurons

Siddharth Jayakumar[1,2], Shlesha Richhariya[1], O Venkateswara Reddy[1], Michael J Texada[3], Gaiti Hasan[1]*

[1]National Centre for Biological Sciences, Tata Institute of Fundamental Research, Bangalore, India; [2]Manipal University, Manipal, India; [3]Janelia Research Campus, Howard Hughes Medical Institute, Ashburn, United States

**Abstract** Neuronal circuits are known to integrate nutritional information, but the identity of the circuit components is not completely understood. Amino acids are a class of nutrients that are vital for the growth and function of an organism. Here, we report a neuronal circuit that allows *Drosophila* larvae to overcome amino acid deprivation and pupariate. We find that nutrient stress is sensed by the class IV multidendritic cholinergic neurons. Through live calcium imaging experiments, we show that these cholinergic stimuli are conveyed to glutamatergic neurons in the ventral ganglion through mAChR. We further show that IP$_3$R-dependent calcium transients in the glutamatergic neurons convey this signal to downstream medial neurosecretory cells (mNSCs). The circuit ultimately converges at the ring gland and regulates expression of ecdysteroid biosynthetic genes. Activity in this circuit is thus likely to be an adaptation that provides a layer of regulation to help surpass nutritional stress during development.

*For correspondence: gaiti@ncbs.res.in

## Introduction

Animals frequently find themselves in situations of nutritional deprivation. To combat these lean periods, physiological mechanisms have evolved that are common to both vertebrates and invertebrates (*Waterson and Horvath, 2015*; *Zhang et al., 2002*). Such mechanisms require the animal to integrate sensory perception of nutrient deprivation with appropriate metabolic changes. The nervous system plays a central role in this process, and communication between multiple neuronal cell types can regulate the necessary metabolic and hormonal changes required for coordinating an organismal response (*Chantranupong et al., 2015*; *Waterson and Horvath, 2015*). The vertebrate hindbrain acts as a central regulator of energy balance. Nuclei of the solitary tract integrate energy status signals from relevant inputs such as blood-borne endocrine signals and synaptic signals from the gastrointestinal tract and peripheral neurons, to modulate appetite and feeding (*Grill and Hayes, 2012*). However, the specific identity and circuitry of neurons responsible for sensing and responding to nutritional cues is not completely understood. This is in part due to the complexity of the vertebrate brain, in which monitoring activity in specific neuronal subtypes is challenging. A less-complex nervous system, consisting of approximately 10,000 neurons (*Scott et al., 2001*), compared with 70 million neurons in the mouse brain (*Economo et al., 2016*), makes *Drosophila* larvae a powerful system to elucidate central brain circuitry underlying systemic responses to nutrient deprivation (*Bjordal et al., 2014*).

Intracellular signaling mechanisms shape neural responses across circuits and contribute greatly to systemic outcome. For example, ghrelin, a gut-derived orexigenic hormone, affects synaptic plasticity under conditions of nutrient deprivation through intracellular signaling involving calcium

**eLife digest** Insect larvae must feed voraciously to accumulate enough nutrients to tide them over the pupal stage of their lifecycle. Unlike larvae, pupae do not feed but instead use their stored energy reserves to fuel their metamorphosis into adults. To maximise their chances of survival, insect larvae must carefully time their transformation into pupae based on both the availability of nutrients in the environment and their own energy stores.

The circuit of neurons within the larval nervous system that detects external nutrient levels, and then relays that information to the insect's metabolic system, remains unknown. This circuit is also of interest because many animal species are thought to use it to slow down their metabolism during periods of food deprivation. Jayakumar et al. therefore set out to identify this circuit by studying how genetically modified fruit fly larvae transform into pupae when nutrients are in short supply.

The experiments show that mutant larvae that lack a protein called $IP_3R$ struggle to turn into pupae when fed a diet deficient in proteins. $IP_3R$ proteins are ion channels that control the release of calcium ions from stores within the cells. Jayakumar et al. showed that food that is deficient in nutrients triggers some larval neurons to release a chemical called acetylcholine, which in turn activates receptors on certain other neurons that communicate using the signalling molecule glutamate. In normal insects, this causes the glutamate-producing neurons to release calcium ions through their $IP_3R$ channels. The calcium ions then activate a chain of events that ultimately causes other cells to produce a hormone called ecdysone, which drives the transformation from larva to pupa. In $IP_3R$ mutants, by contrast, the absence of calcium ion release keeps the insect in the larval stage.

This circuit helps to explain how some insects and other animals are able to survive being deprived of food for extended periods. Further work will be required to understand how a lack of protein in the diet changes the signalling properties of cells in various parts of the circuit.

(*Yang et al., 2011*). Nonetheless, intracellular signaling pathways responsible for synaptic plasticity in circuits that regulate organismal responses to an altered nutrient status need further elucidation. Intracellular calcium signaling evolved in parallel with multi-cellularity (*Cai, 2008*), and may therefore function in coordinating systemic metabolic responses (*Chantranupong et al., 2015*) of metazoans. A key component of intracellular calcium signaling is the Inositol 1, 4, 5-trisphosphate receptor ($IP_3R$). These are calcium channels that mediate intracellular calcium release from the endoplasmic reticulum (ER) in response to extracellular stimuli (*Streb et al., 1983*). In vertebrates, calcium release through $IP_3R2$ and $IP_3R3$ is required in various classes of non-excitable cells for metabolic control (*Wang et al., 2012*) and exocrine secretion of insulin or amylase from the pancreas (*Berggren et al., 2004*; *Futatsugi et al., 2005*). $IP_3R1$ is expressed in different classes of neurons where it regulates processes ranging from synaptic plasticity (*Nishiyama et al., 2000*) to axonal guidance (*Xiang et al., 2002*). Due to the broad expression of most components of metazoan intracellular signaling, including the $IP_3R$ family, identifying cell-specific function in vivo can be challenging. *Drosophila* genetics provides the tools for such cell-specific analysis.

In *Drosophila*, $IP_3R$ is encoded by the single *itpr* gene (*Hasan and Rosbash, 1992*). $IP_3R$ mutants exhibit delayed moulting (*Venkatesh and Hasan, 1997*) recently attributed to release of the steroid hormone ecdysone from the prothoracic gland (*Yamanaka et al., 2015*). While null alleles are lethal as second instar larvae, heteroallelic hypomorphs exhibit developmental and metabolic phenotypes. These range from lethality across larval stages to hyperphagic adults with altered lipid metabolism. The focus of adult metabolic defects observed in *itpr* mutants appears to be the central nervous system (*Subramanian et al., 2013a*, *2013b*). For a better understanding of neuronal $IP_3R$ function in the context of metabolic regulation, we chose to study the *Drosophila* larval to pupal transition. This transition requires systemic integration of the nutritional state of late-stage larvae with release of hormones that drive pupariation (*Andersen et al., 2013*). Here, we identify a neural circuit that allows *Drosophila* larvae to overcome chronic protein-deprivation and pupariate. We demonstrate that nutrient sensitive plasticity of this circuit requires intracellular calcium signaling in newly identified glutamatergic neurons of the ventral ganglion.

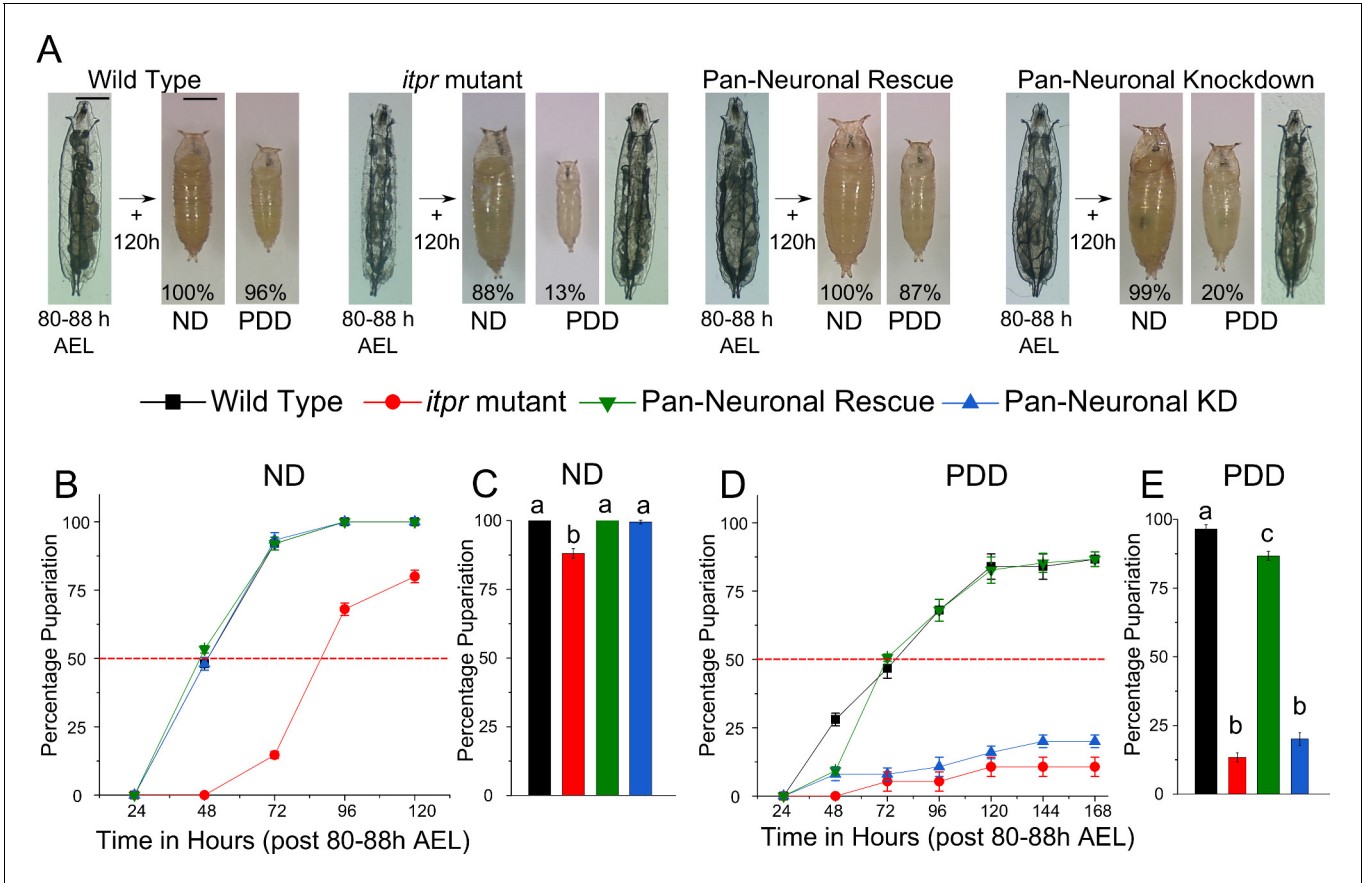

**Figure 1.** Pupariation in a protein-deprived environment requires intracellular calcium signaling in neurons. (A) Representative images of larvae and pupae of indicated genotypes subjected to either protein-deprived diet (PDD) or normal diet (ND). Percentages refer to pupariation. B and D Percentage pupariation of indicated genotypes represented as mean ± SEM over hours after transfer to the indicated media at 80–88 hr after egg laying (AEL). Dotted lines indicate 50% viability. C and E Bars represent mean percentage pupariation at 120 hr (± SEM). Larvae were transferred to the indicated diet at 80–88 hr AEL. All pupariation experiments were performed with N ≥ 6 batches, with 25 larvae in each batch. Bars with the same alphabet represent statistically indistinguishable groups (one-way ANOVA with a post hoc Tukey's test p<0.05).

The following figure supplement is available for figure 1:

**Figure supplement 1.** Pupariation under protein-deprivation requires neuronal IP$_3$R.

## Results

### Pupariation in a protein-deprived environment requires intracellular calcium signaling in neurons

To assess the role of IP$_3$R in nutrient stress, *itpr* mutants (*itpr^{ka1091/ug3}*) were transferred as early third-instar larvae from a normal diet (ND) to a protein-deprived diet (PDD) containing only sucrose (*Figure 1A* and *Figure 1—figure supplement 1A and B*). At 120 hr post transfer, wild-type larvae exhibited complete pupariation on either ND or PDD, whereas *itpr* mutant larvae exhibited a decrease in pupariation on ND which was worsened significantly on PDD (*Figure 1A*). Pupariation in wild-type larvae was delayed slightly when subjected to PDD, whereas *itpr* mutants, which barely pupate on PDD, were delayed considerably even on a normal diet (*Figure 1B,C,D and E*). Protein is the likely nutritional cue necessary for pupariation by *itpr* mutants, because the extent of pupariation on a lipid-deprived diet was significantly higher than on PDD and closer to pupariation on a normal diet (*Figure 1—figure supplement 1C*). Moreover, pupariation in *itpr* mutants was restored when the PDD was supplemented with amino acids and vitamins (*Figure 1—figure supplement 1D*). Interestingly, *itpr* mutant larvae feed in excess of controls (*Figure 1—figure supplement 1E*), but their

body weights are similar to that of wild-type larvae at the time of transfer to the PDD, indicating that excess feeding may be an attempt to compensate for metabolic changes due to altered intracellular calcium signaling (*Figure 1—figure supplement 1F*). Pan-neuronal restoration of *itpr* function in the mutant by expressing wild-type *itpr* cDNA (*itpr+*) with *elav-GAL4* rescued the pupariation deficit, whereas pan-neuronal knockdown of the IP$_3$R mimicked the *itpr* mutant phenotype on PDD, while exhibiting complete pupariation on ND (*Figure 1A,B,C,D and E*). These data demonstrate that pupariation in a protein-deprived condition requires intracellular calcium signaling through the IP$_3$R in neurons.

## Protein-deprived larvae require IP$_3$R function in glutamatergic neurons for pupariation

To identify neuronal subsets that require *itpr* function for pupariation in protein-deprived conditions, IP$_3$R knockdown was performed in various neuronal subsets. Among the tested subsets, the strongest pupariation deficit was observed upon IP$_3$R knockdown in glutamatergic neurons (*Figure 2—figure supplement 1A*). Peptidergic neurons also showed a significant pupariation deficit, but this effect was not diet specific (*Figure 2—figure supplement 1A* and Table 2). Interestingly, when pan-neuronal knockdown of the IP$_3$R was restricted to the brain lobes by introducing *tsh-GAL80*, pupariation was restored from less than 25% (*Figure 1E*) to more than 75% (*Figure 2—figure supplement 1B*). These data indicate that IP$_3$R function is required to a greater extent in neurons located in the ventral ganglion for pupariation on PDD as compared with neurons located in the central brain lobes. We therefore used two independent *GAL4* strains, *vglut^VGN6341^* and *vglut^VGN9281(2)^* (subsequently referred to as *VGN6341* and *VGN9281-2*), with contrasting expression patterns within the glutamatergic population of the ventral ganglion (*Figure 2A*), to delimit the subpopulation required for pupariation on PDD. The *VGN6341* and *VGN9281-2* strains differ in their expression patterns especially in the third thoracic and initial abdominal segments of the ventral ganglion (*Figure 2A*). IP$_3$R knockdown with *VGN6341* resulted in significantly reduced pupariation on PDD, whereas larvae with IP$_3$R knockdown in *VGN9281-2*-expressing cells pupariated similar to control larvae (*Figure 2B*). Normal pupariation was observed when IP$_3$R knockdown was restricted by introducing *tsh-GAL80* in the background of *VGN6341* (referred to as restricted *VGN6341*; *Figure 2B*). *tsh-GAL80* restricted expression of *VGN6341*, to the central brain and a few neurons in the ventral ganglion. In a complementary experiment, pupariation in the *itpr* mutant was restored by expression of *UAS-itpr+* in *VGN6341* expressing neurons but not with the GAL80-restricted *VGN6341* neurons (*Figure 2C*). Comparison of *VGN9281-2* and *VGN6341* expression with and without *tsh-GAL80* pointed to a region between the third thoracic and fifth abdominal segments in the ventral ganglion, subsequently designated as the mid-ventral ganglion or mVG, where IP$_3$R function is required for pupariation on PDD (*Figure 2D*). These *VGN6341*-expressing cells were confirmed as glutamatergic by co-immunostaining of *VGN6341* driven GFP with DvGlut, an established marker of glutamatergic neurons (*Daniels et al., 2004*) (*Figure 2E*, *Figure 2—figure supplement 1C*). Although a role for peripheral glutamatergic neurons of the mVG remains possible, we focussed on the central ones as they were more easily and repeatedly identifiable.

## Glutamatergic neurons of the mVG respond to cholinergic stimuli in a diet-dependent manner

To identify GPCRs that function in glutamatergic neurons and stimulate the IP$_3$R under protein-deficient conditions, a genetic RNAi (IR) screen was performed with *VGN6341* and all publicly available RNAi strains for GPCRs in the *Drosophila* genome (*Figure 3—figure supplement 1*, *Supplementary file 3*). Twelve GPCRs identified in the screen (*Table 1*) were validated as functioning upstream of the IP$_3$R, by rescue of pupariation with two components of the IP$_3$ signaling pathway, a constitutively active Gαq transgene *AcGq* and *dSTIM* (*Table 1*). Among the identified GPCRs, we tested further the role of the muscarinic Acetylcholine Receptor (Flybase *mAChR-A*, *mAChR-60C*, *CG4356* and henceforth, *mAChR*) in pupariation. The mAChR activates Gq/PLCβ signaling leading to IP$_3$ production and IP$_3$-mediated Ca$^{2+}$ release in *Drosophila* cells (*Millar et al., 1995*; *Srikanth et al., 2006*). Knockdown of the *mAChR* in glutamatergic neurons of the ventral ganglion significantly reduced pupariation on protein-deficient media and expression of the *mAChR+* transgene in the same neuronal subset in the *itpr* mutant rescued pupariation (*Figure 3A*). These data

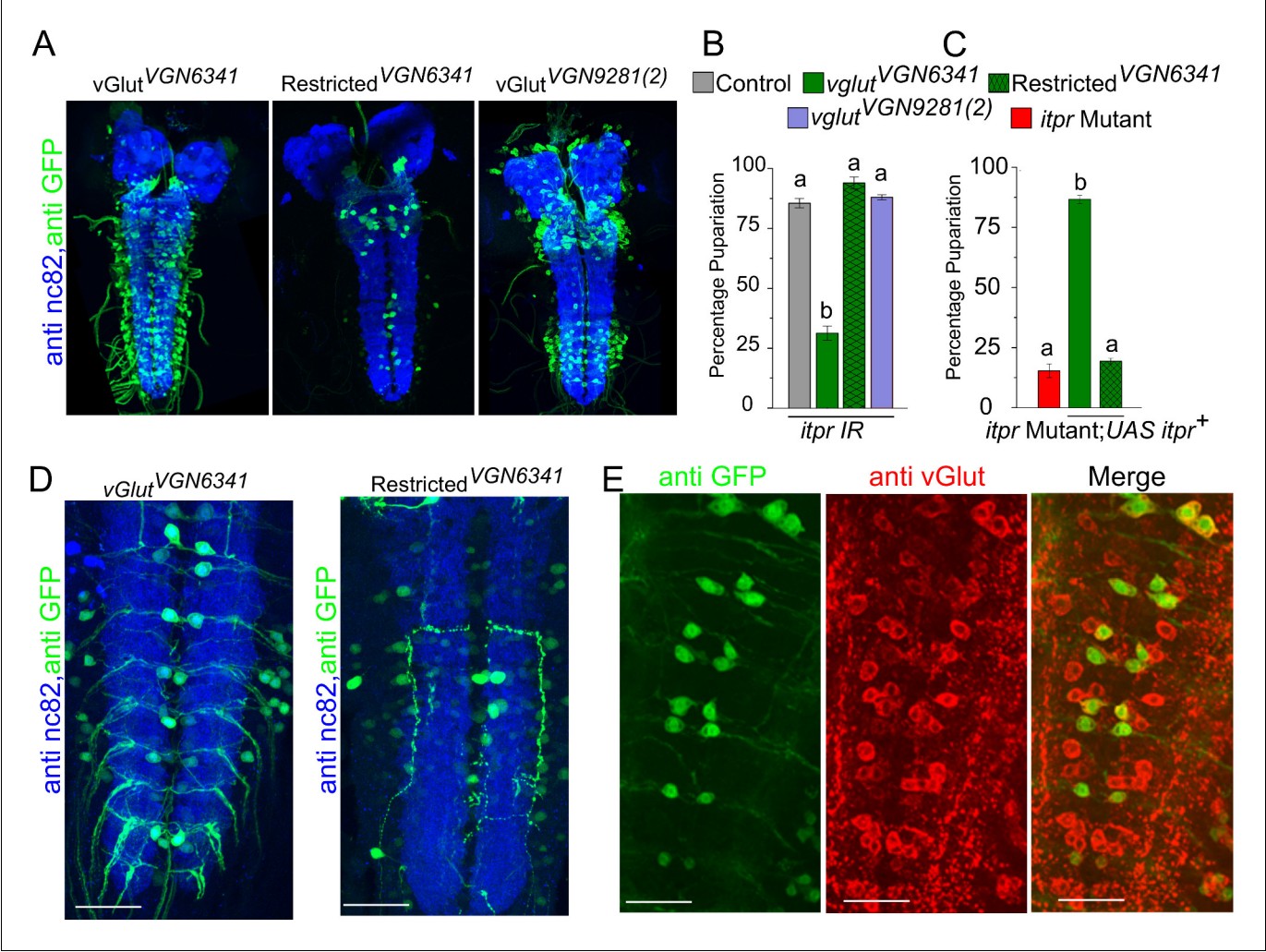

**Figure 2.** Knockdown of the IP$_3$R in glutamatergic neurons prevents pupariation upon PDD. (**A**) Expression patterns of *GAL4* drivers used in (**B**) determined using *UAS-eGFP* and co-stained with anti-nc82. (**B** and **C**) Bars show mean percentage pupariation (± SEM) of the indicated genotypes on PDD. N ≥ 6 batches with 25 larvae each. (**D**) Images of selected substacks of the ventral ganglion of *VGN6341-GAL4*, with and without *tsh-GAL80*, expressing *UAS-eGFP*, double labelled with anti-nc82. (**E**) Selected substacks showing overlap of all *dvGlut*-positive cells and GFP-positive cells marked by *VGN6341-GAL4* in the ventral ganglion. Scale bars indicate 50 µm. Bars with the same alphabet represent statistically indistinguishable groups (one-way ANOVA with a post hoc Tukey's test p<0.05).

The following figure supplement is available for figure 2:

**Figure supplement 1.** Knockdown of the IP$_3$R in glutamatergic neurons prevents pupariation upon protein-deprivation.

support mAChR stimulation followed by Ca$^{2+}$ release through the IP$_3$R as a mechanism required by glutamatergic neurons of the mVG for pupariation under conditions of nutrient stress.

Next we assessed the response of *VGN6341*-expressing neurons in the mVG to carbamylcholine (CCh), an mAChR agonist (*Offermanns et al., 1994*). We measured CCh responses in ex vivo preparations of third-instar larval brains (*Figure 3B*) that were subjected to either ND or PDD for 18 hr, by *VGN6341*-driven expression of the genetically encoded calcium indicator *GCaMP6m*. Upon stimulation with CCh, calcium transients were observed in multiple *VGN6341* marked cells from larvae fed on a normal diet (*Figure 3C,D* and *Figure 3—figure supplement 2C*). The specificity of this response was ascertained by pre-incubation with atropine, an established and specific antagonist of the mAChR (*Fryer and Maclagan, 1984*), which abolished the response (*Figure 3—figure supplement 2A and B*). Not surprisingly, the response to CCh was significantly attenuated in *itpr* mutants

**Table 1.** Validated Hits from the GPCR RNAi Screen. Percentage pupariation (rounded to the nearest integer) upon GPCR knockdown performed with *VGN6341-GAL4* as well as their rescue by overexpression of *dSTIM* and constitutively active form of *Gq* on PDD.

| Sl. no. | RNAi line (CG) | Receptor category | Receptor | Larvae to pupae with RNAi (%) | Larvae to pupae rescue with UAS-dSTIM (%) | Larvae to pupae rescue with UAS-AcGq (%) |
|---|---|---|---|---|---|---|
| 1 | 16785 | Frizzled | Frizzled 3 | 0 | 80 | 44 |
| 2 | 8784 | NPD | Pyrokinin 2 receptor 1 | 16 | 90 | 84 |
| 3 | 7395 | NPD | sNPF receptor | 16 | 84 | 80 |
| 4 | 8795 | NPD | Pyrokinin 2 receptor 2 | 20 | 72 | 56 |
| 5 | 2114 | NPD | Fmrf receptor | 20 | 88 | 44 |
| 6 | 14593 | NPD | CCHamide-2 receptor | 20 | 68 | 56 |
| 7 | 4356 | Acetylcholine | Muscarinic acetylcholine receptor at 60C | 20 | 88 | 80 |
| 8 | 6515 | NPD | Tachykinin-like receptor at 86C | 20 | 68 | 56 |
| 9 | 16766 | Monoamines | Tyramine receptor II | 24 | 92 | 76 |
| 10 | 15274 (Earlier 33310) | GABA, Glutamate | Metabotropic GABA-B receptor subtype 1 | 28 | 76 | 72 |
| 11 | 10823 | NPD | SIFa receptor | 28 | 78 | 88 |
| 12 | 10001 | NPD | Allatostatin receptor | 44 | 92 | 76 |

on a normal diet (*Figure 3C,D* and *Figure 3—figure supplement 2C*). On PDD, the response to CCh was reduced in neurons from control larvae, and it was nearly absent in neurons from *itpr* mutants (*Figure 3C,E,F and G*; *Videos 1* and *2*). To assess whether there was a temporal component to these responses, we measured responses at either 2 hr or 18 hr on NDD and PDD. In control larvae, the CCh response of glutamatergic neurons reduced significantly from 2 hr to 18 hr on PDD (*Figure 3—figure supplement 2E*). The strongly attenuated CCh response observed in *itpr* mutants at 18 hr on PDD was not evident at 2 hr on PDD (*Figure 3—figure supplement 2E*). These data indicate that a physiological change in the mVG glutamatergic neurons happens on PDD over time. The change in response to CCh over time was not evident on ND in either control or *itpr* mutant neurons (*Figure 3—figure supplement 2D*). Importantly, the response of glutamatergic neurons in *itpr* mutants could be rescued by supplementing PDD with essential amino acids (EAA) (*Figure 3E*).

**Table 2.** Percentage pupariation on ND. Percentage pupariation (rounded to the nearest integer) on ND of indicated genotypes.

| Sl. no. | Genotype | Percentage pupariation on ND |
|---|---|---|
| 1 | OK371-GAL4>itpr IR | 97 |
| 2 | VGN6341-GAL4>itpr IR | 95 |
| 3 | ChAT-GAL4>itpr IR | 99 |
| 4 | Dimm-GAL4>itpr IR | 63 |
| 5 | Restricted *dimm-GAL4>itpr IR* | 86 |
| 6 | Ppk-GAL4>itpr IR | 100 |
| 7 | Restricted VGN6341-GAL4>itpr IR | 97 |
| 8 | VGN6341-GAL4>mAChR IR | 99 |
| 9 | Restricted VGN6341-GAL4>mAChR IR | 100 |
| 10 | Dimm-GAL4>mAChR IR | 97 |
| 11 | Dimm-GAL4>mGluR$_A$ IR | 83 |

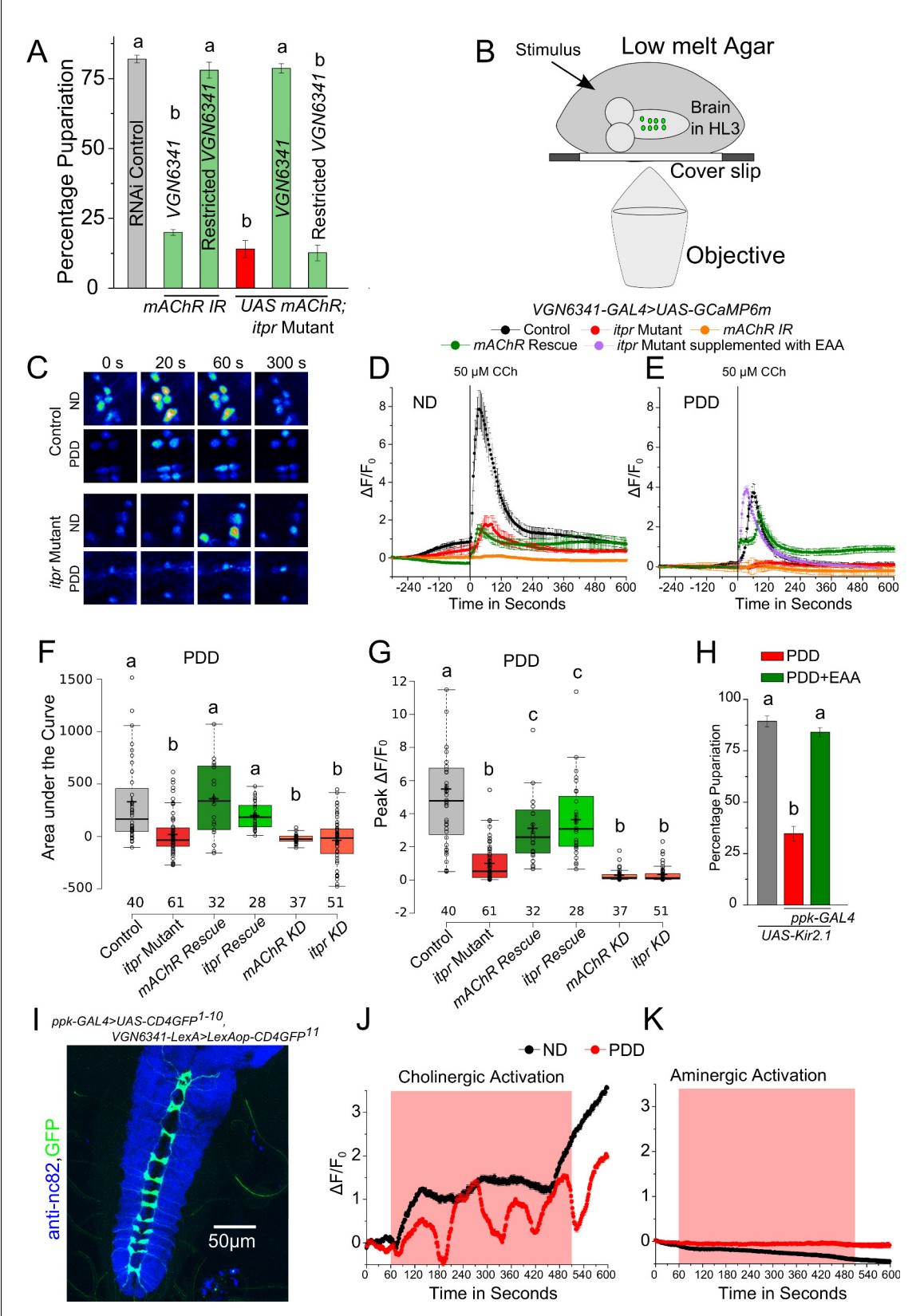

**Figure 3.** Cholinergic inputs convey nutrient-stress signals to glutamatergic neurons of the ventral ganglion. (**A**) Bars indicate mean percentage pupariation (± SEM) of indicated genotypes subjected to PDD. N ≥ 6 batches with 25 larvae each. (**B**) Schematic illustrating the setup used to image

*Figure 3 continued on next page*

*Figure 3 continued*

neurons of interest from the larval ventral ganglion. (**C**) Representative images showing calcium activity measured by GCaMP6m in the mVG neurons of indicated genotypes at indicated time points from a time series. (**D** and **E**) Traces represent time series of the mean normalized GCaMP6m responses (± SEM) from the mVG neurons of the indicated genotypes upon stimulation with carbamylcholine (CCh). (**F**) and (**G**) Box plots represent Area under the Curve (**F**) and Peak change in fluorescence (**G**) quantified from (**E**). In box plots, center lines show the medians; box limits indicate the 25th and 75th percentiles, whiskers extend 1.5 times the interquartile range from the 25th and 75th percentiles, open circles represent each data point and numbers below represent total number of cells measured. (**H**) Bars indicate mean percentage pupariation (± SEM) of the indicated genotypes on PDD. N ≥ 6 batches with 25 larvae each. (**I**) Image showing GRASP between *pickpocket-GAL4* and *VGN6341-LexA*. **J** and **K** Traces represent time series of mean normalized GCaMP6m responses (± SEM) from mVG glutamatergic cells upon optogenetic activation of either cholinergic (**J**) or aminergic (**K**) domains. Grey box indicates duration of optogenetic activation. Bars with the same alphabet represent statistically indistinguishable groups (one-way ANOVA with a post hoc Tukey's test p<0.05).

The following figure supplements are available for figure 3:

**Figure supplement 1.** Identification of GPCRs that stimulate IP$_3$-mediated calcium signaling in response to PDD.

**Figure supplement 2.** Cholinergic neurons are important for responding to nutrient stress.

**Figure supplement 3.** *ppk* class IV multidendritic neurons activate *VGN6341* marked glutamatergic interneurons.

Thus, loss of dietary EAA appears to be a cause for abrogation of the CCh response in *itpr* mutant glutamatergic neurons after 18 hr on PDD. The response to CCh was also absent in glutamatergic neurons of the mVG with knockdown of either *mAChR* or *itpr* on ND (**Figure 3D** and **Figure 3—figure supplement 2C**) and PDD (**Figure 3E,F,G**). The absence of the CCh response on ND in *itpr* and *mAChR* knockdowns, even though validating the original response to be through the IP$_3$R, did not correlate with a defect in pupariation (Table 2), suggesting that mAChR and IP$_3$R function in mVG neurons is critically required on PDD but not relevant on ND (see discussion). The response to CCh on PDD was rescued to a significant extent by over-expression of either *mAChR$^+$* or *itpr$^+$* in *VGN6341*-marked neurons of *itpr* mutants (**Figure 3E,F and G**), and correlated with rescue of pupariation observed in these genotypes (**Figures 2C** and **3A**). These data suggest that cholinergic inputs convey protein-starvation to *VGN6341*-marked glutamatergic neurons possibly through acetylcholine, that activate mAChR and the IP$_3$R for pupariation.

To test if cholinergic stimuli signal protein-deprivation, recycling of synaptic vesicles was blocked in cholinergic neurons by expression of the dynamin mutant transgene *Shibire$^{ts}$ (UAS-Shi$^{ts1}$)* with the cholinergic driver *ChaT-GAL4*. Pupariation was reduced significantly when cholinergic transmission was blocked, by transferring larvae (80–88 hr AEL) to the restrictive temperature (29°C) for 48 hr (till 128–136 hr AEL), concurrent with protein-deprivation (**Figure 3—figure supplement 2G**). The same experiment performed either with larvae on ND or with larvae on PDD but at the permissive temperature (22°C) supported pupariation (**Figure 3—figure supplement 2F and H**). When *Shi$^{ts}$* expression was driven by a combination of *ChaT-GAL4* and *ppk-GAL80* (blocking GAL4 activity in multidendritic sensory neurons),

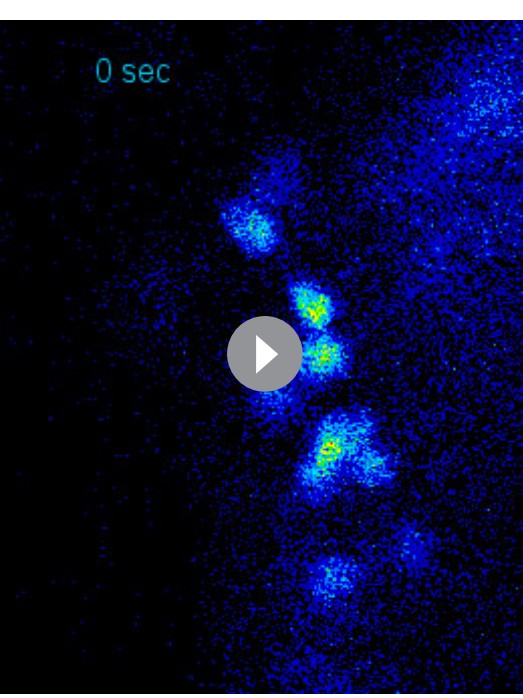

**Video 1.** Response to CCh in *VGN 6341* neurons of the mVG from control larvae on PDD. The green flash indicates point of point of addition of CCh.

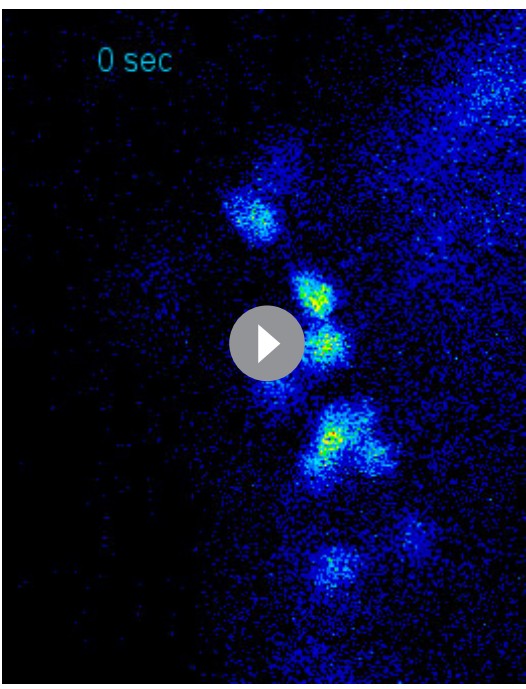

**Video 2.** Response to CCh in *VGN 6341* neurons of the mVG from *itpr* mutant larvae on PDD. Green flash indicates point of addition of CCh.

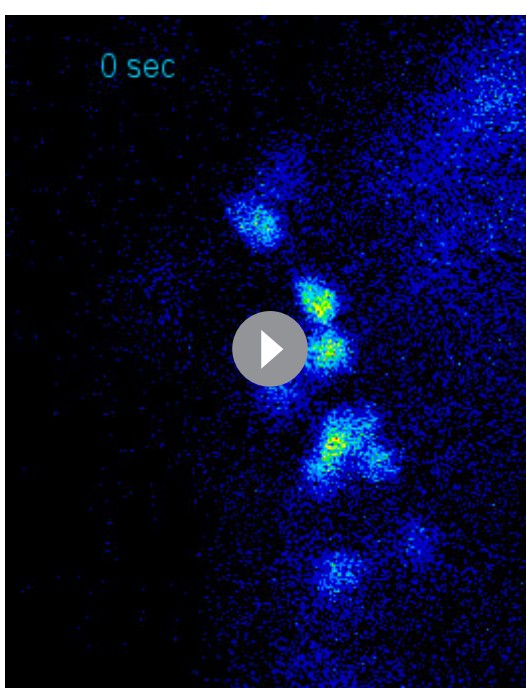

**Video 3.** Calcium transients observed in glutamatergic neurons of the mVG as a result of optogenetic activation of cholinergic neurons.

normal pupariation was observed on the protein-deficient diet (*Figure 3—figure supplement 2G and H*), suggesting the requirement of neurons expressing the Pickpocket (ppk) channel (*Adams et al., 1998*) in sensing protein-deficient conditions prior to pupariation. This idea was tested directly by expression of *Shi^ts* or a hyper-polarising potassium channel Kir2.1 in neurons expressing Pickpocket (*ppk-GAL4*), both of which caused a severe pupariation deficit (*Figure 3H*, *Figure 3—figure supplement 2G*). Pupariation was restored in these animals by supplementing the protein-deficient diet with a mixture of EAA (*Figure 3H*). Activity in *ppk-GAL4*-marked neurons is thus required for pupariation on PDD but not on ND. In contrast, pupariation was unaffected when activity was inhibited in cholinergic neurons marked by *19-12-GAL4* expression, which does not overlap with *ppk-GAL4* (*Figure 3—figure supplement 2G and H*; *Yan et al., 2013*).

The presence of direct cholinergic inputs to *VGN6341*-marked glutamatergic neurons was tested next by performing an experiment for genetic reconstitution across synaptic partners (GRASP) (*Feinberg et al., 2008*). GRASP signals between *VGN6341*- and *ppk*-marked neurons were detected in the neuropil of the ventral ganglion (*Figure 3I*), which is a synaptically dense region along the midline. GRASP constructs expressing the split GFP components individually with either *ppk-GAL4* or *VGN6341-LexA* had no GFP expression (*Figure 3—figure supplement 3A and B*). To test if the observed connections are functional, cholinergic neurons were marked by *ChaT-LexA* and optogenetically activated with a red-shifted channelrhodopsin variant, *LexAop-CsChrimson*. Calcium transients were observed in glutamatergic cells of the mVG simultaneously with optogenetic activation of cholinergic neurons on either ND or PDD (*Figure 3J*; *Video 3*). On PDD, interestingly, the transients appeared to oscillate. As a control, we tested optogenetic activation of aminergic neurons (marked by *HL9-LexA*). This did not elicit a response in glutamatergic neurons of the mVG (*Figure 3K*). To confirm that activation of VGN6341 neurons was through inputs from ppk neurons, we performed an optogenetic activation experiment with *TrpA1-QF* which marks class IV multidendritic neurons (*Petersen and Stowers, 2011*; subsequently referred to as *ppk-QF*). A robust activation of *VGN6341*-marked cells expressing the red shifted calcium indicator jRCaMP1b was observed upon optogenetic activation of ppk neurons expressing *QUAS-ChR2* (*Figure 3—*

*figure supplement 3*). Knockdown of the IP$_3$R in cholinergic neurons did not change pupariation on PDD (*Figure 2—figure supplement 1A*), and restoring IP$_3$R function in cholinergic neurons failed to rescue the *itpr* mutant (*Figure 3—figure supplement 2I*). As normal pupariation was observed in animals with cholinergic knockdown of the IP$_3$R on PDD, we did not test cholinergic stimulation of glutamatergic neurons in IP$_3$R mutants. Instead, to understand the basis of the pupariation defect on PDD, we investigated next the postsynaptic partners of glutamatergic neurons marked by *VGN6341-GAL4*.

## Glutamatergic neurons of the mVG convey nutrient stress to medial neurosecretory cells in the brain

Neuropeptides are known to modulate organismal responses to changes in diet in vertebrates as well as insects (*Morton et al., 2014*; *Nässel and Winther, 2010*). To test, whether the mVG interneurons synapse on to peptidergic cells, we imaged the two domains at high resolution. Anterior glutamatergic projections arising from near the central mVG interneurons (*Figure 4A and B*, arrow heads) reach posterior projections of the mNSCs (*Figure 4A and B*, asterisks). These anterior glutamatergic projections appear to originate from the central mVG neurons, extend laterally toward the midline for a short distance, and then project to the anterior (*Figure 4D* and *video 4*). They do not arise from peripheral mVG neurons that appear to project solely to the midline neuropil of the VG. Similar connectivity was observed upon marking the mVG glutamatergic neurons and specifically the mNSCs (*Figure 4C*).

Additionally, a GRASP experiment was performed between *VGN6341-LexA*-marked neurons and peptidergic neurons marked by *dimm-GAL4* (*Figure 5A*). The ensuing GRASP presented a complex pattern comprising medial Neurosecretory Cells (mNSCs) in the central brain, multiple projections between the mVGs and the mNSCs, and a few cell bodies and projections in the periphery of the ventral ganglion (*Figure 5A* and *Figure 5—figure supplement 1A*). Controls with either GAL4 or LexA driving individual GRASP constructs elicited no GFP immunostaining (*Figure 5—figure supplement 2A and B*). To test the functional significance of this apparent connectivity between peptidergic and glutamatergic cells, we measured pupariation on PDD after knockdown of various glutamatergic receptors in *dimm*-positive neurons. Amongst the five receptors tested, pupariation was reduced significantly in animals with knockdown of the metabotropic glutamate receptor A (mGluR$_A$; *Figure 5B* and *Figure 5—figure supplement 1B*). Knockdown of *mGluR$_A$* with two independent RNAi strains showed differing but significant deficits in pupariation (10% and 40%; *Figure 5B*, *Figure 5—figure supplement 1B*). We attribute this difference to the strength of the RNAi knockdown in peptidergic cells from the two different RNAi strains. Knockdown of other glutamate receptor classes resulted in normal pupariation (*Figure 5—figure supplement 1B*). Restriction of *mGluR$_A$* knockdown in peptidergic neurons of the central brain and in Dilp2-positive mNSCs (*Dilp2-GAL4)* also resulted in pupariation deficits on PDD (*Figure 5B*). In contrast, normal pupariation was observed by restricting knockdown of IP$_3$R to *VGN6341*-marked glutamatergic neurons of the central brain (*Figure 2B*). Taken together these data support the innervation of central brain peptidergic neurons, specifically the mNSCs, by glutamatergic neurons of the ventral ganglion. This innervation appears relevant for pupariation on PDD.

mNSCs in *Drosophila* central brain secrete multiple neuropeptides, including insulin-like peptides (Dilps), which regulate the response of an organism to dietary changes (*Dus et al., 2015*). Upon co-immunostaining with an antibody against Dilp2 (*Géminard et al., 2009*), five of the seven Dilp2-positive cells in the mNSCs were 'GRASPed' by *VGN6341* (*Figure 5C*) as suggested earlier by immunostaining of mVG cells and the mNSCs (*Figure 4C*). The presence of dendritic fields close to the mNSC soma has been described earlier (*Nässel et al., 2008*; *Vallejo et al., 2015*) and was confirmed by simultaneous expression of an axonal and a dendritic marker with *Dilp2-GAL4* (*Figure 5D*). Thus, axonal projections from *VGN6341*-marked glutamatergic neurons probably stimulate a subset of mNSCs. The presence of functional synaptic connections between *VGN6341-GAL4* and mNSCs was tested next. *CsChrimson*-expressing *VGN6341* cells were optogenetically activated in ex vivo preparations in which *GCaMP6m* was expressed in peptidergic cells, including the mNSCs, marked by *dimm-LexA::p65*. Upon optogenetic stimulation of *CsChrimson* in *VGN6341* marked neurons calcium transients were observed in the mNSCs of larvae from both normal and protein-deficient diets (*Figure 5E,F,G and H*; *Video 5*). Similar activation of the mNSCs was obtained when *VGN6341* marked neurons were thermogenically activated with *dTrpA1*, a temperature-activated

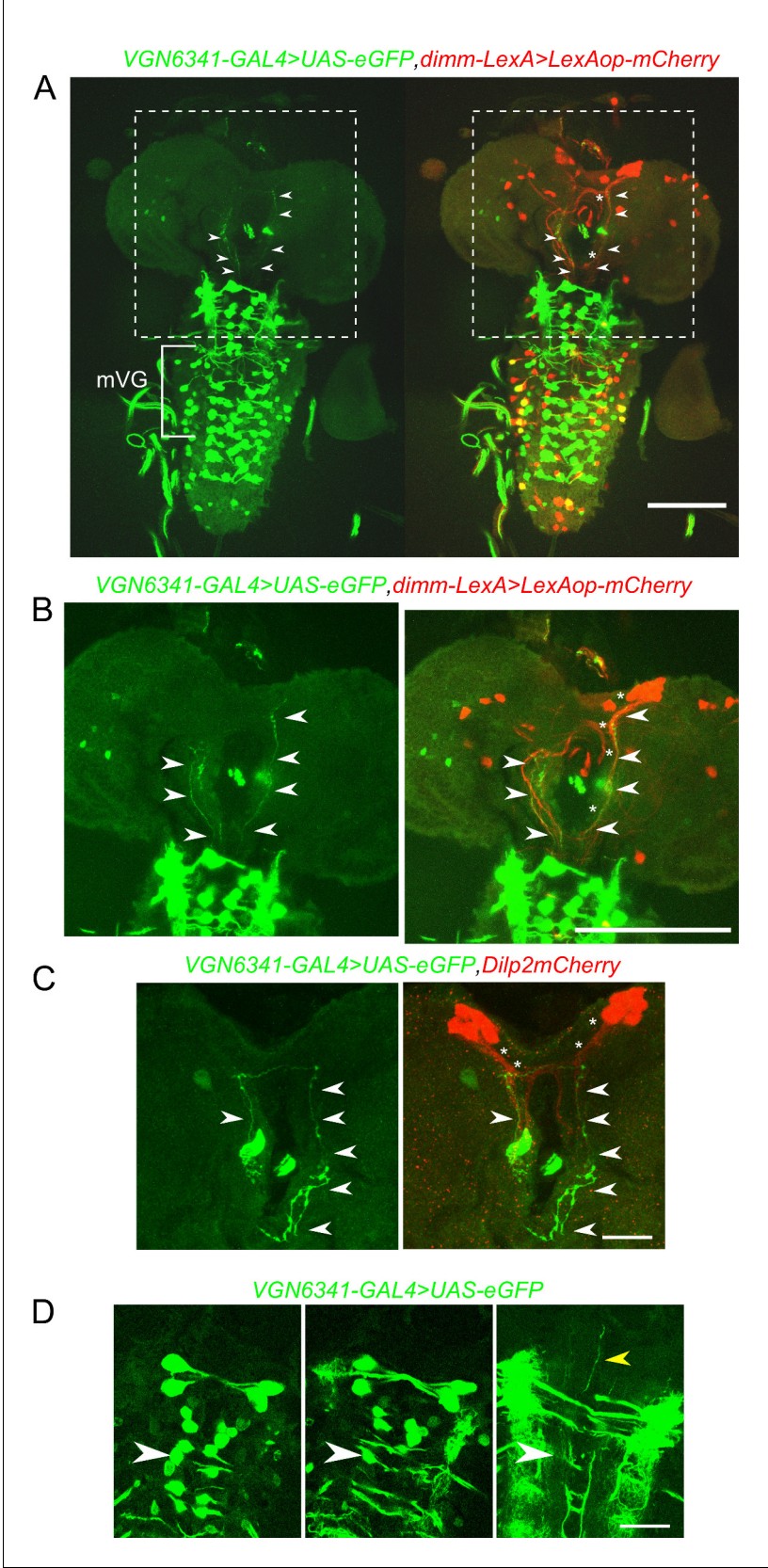

**Figure 4.** Glutamatergic neurons in the larval ventral ganglion project to peptidergic neurons in the mNSC. **A** and **B** Selected confocal stacks showing the neurites marked by *VGN6341-GAL4* driven *UAS-eGFP* (green) and their

*Figure 4 continued*
merged patterns with *dimm-LexA*-driven expression of *LexAop-mCherry* (red). The boxed area in **A** is shown in **B** as a high-magnification image. Arrow heads indicate *VGN6341-GAL4* expressing neurites projecting toward the mNSCs. Asterisks mark *dimm-LexA* labelled projections. (**C**) Neurites marked by *VGN6341-GAL4*-driven eGFP (arrow heads) overlap with projections of the mNSCs marked by Dilp2mCherry (asterisks). (**D**) Selected high-magnification confocal images of *VGN6341-GAL4* driven *UAS-eGFP* with an anterior projecting neurite from a midline mVG neuron. The white arrow head marks the same co-ordinates in all three images. The yellow arrow head shows the ascending projections. Scale bars represent 50 µm in **A** and **B** and 10 µm in **C** and **D**.

cation channel (*Figure 5—figure supplement 1C and E*). Activation of *VGN6341* marked neurons in the *itpr* mutant, however, did not evoke transients either on ND (*Figure 5E and G*) or on PDD (*Figure 5F and H*). The underlying basis for this defect appears to be an inability of the *VGN6341*-marked neurons to stimulate the mNSCs in *itpr* mutants, because their optogenetic self-activation evoked robust calcium transients in *itpr* mutants on ND and PDD (*Figure 5—figure supplement F and G*). Activation of the Restricted *VGN6341-GAL4*, where GAL4 expression is absent from the mVG region, did not elicit a signal from the mNSCs (*Figure 5—figure supplement 2C*).

Next, we tested if artificial activation of glutamatergic neurons marked by *VGN6341* compensated for reduced *itpr* function in the context of pupariation on PDD. NaChBac is a bacterial sodium channel that increases excitability of *Drosophila* neurons (*Nitabach et al., 2006*). Expression of either *dTrpA1, NaChBac,* or *CsChrimson* in glutamatergic neurons marked by *VGN6341* indeed rescued the pupariation defect of the *itpr* mutant on PDD (*Figure 5I*). For rescue experiments with either dTrpA1 or CsChrimson, activation was for 48 hr and this period was concurrent with transfer to PDD from 80–88 hr to 128–136 hr AEL. Taken together, these results suggest that glutamatergic interneurons in the larval mVG receive cholinergic signals indicating absence of dietary amino acids, process these signals in an IP$_3$R-dependent manner, and convey this information to peptidergic cells in the mNSCs for mounting a suitable physiological response to enable pupariation.

## Glutamatergic neurons of the mVG regulate peptide release from mNSC

The effect of CCh-induced calcium transients on neuropeptide release in the mNSCs was tested next. For this purpose, an ex vivo preparation was taken in which cells marked by *dimm-LexA::p65* expressed the rat atrial natriuretic peptide fused with GFP (ANF::GFP) under LexAop control. Peptide release in *Drosophila* neurons has been studied previously by measuring release of *ANF::GFP* (*Shakiryanova et al., 2006*). Upon stimulation with 50 µM CCh, a decay in GFP fluorescence was observed in the mNSCs (*Figure 6A*; *Video 6*). Application of saline did not show a change in fluorescence (*Figure 6A*). The extent of release observed was higher in brains from larvae on PDD as compared to larvae on ND (*Figure 6A*). Release of ANF::GFP was significantly attenuated in mNSCs of the *itpr* mutant and importantly, this could be rescued by expressing *mAChR+* in glutamatergic neurons marked by *VGN6341* (*Figure 6B,C and F*). Moreover, peptide release from the mNSCs was reduced significantly upon knockdown of *mAChR* in *VGN6341*-marked glutamatergic

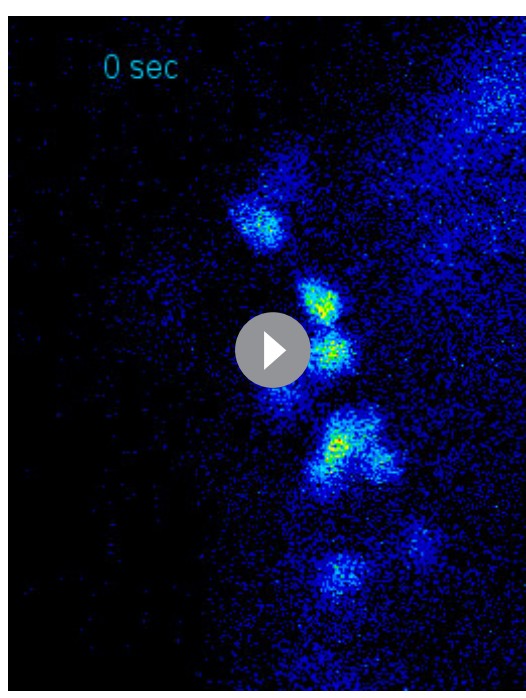

**Video 4.** Projections from *VGN 6341*-marked glutamatergic neurons to the mNSCs.

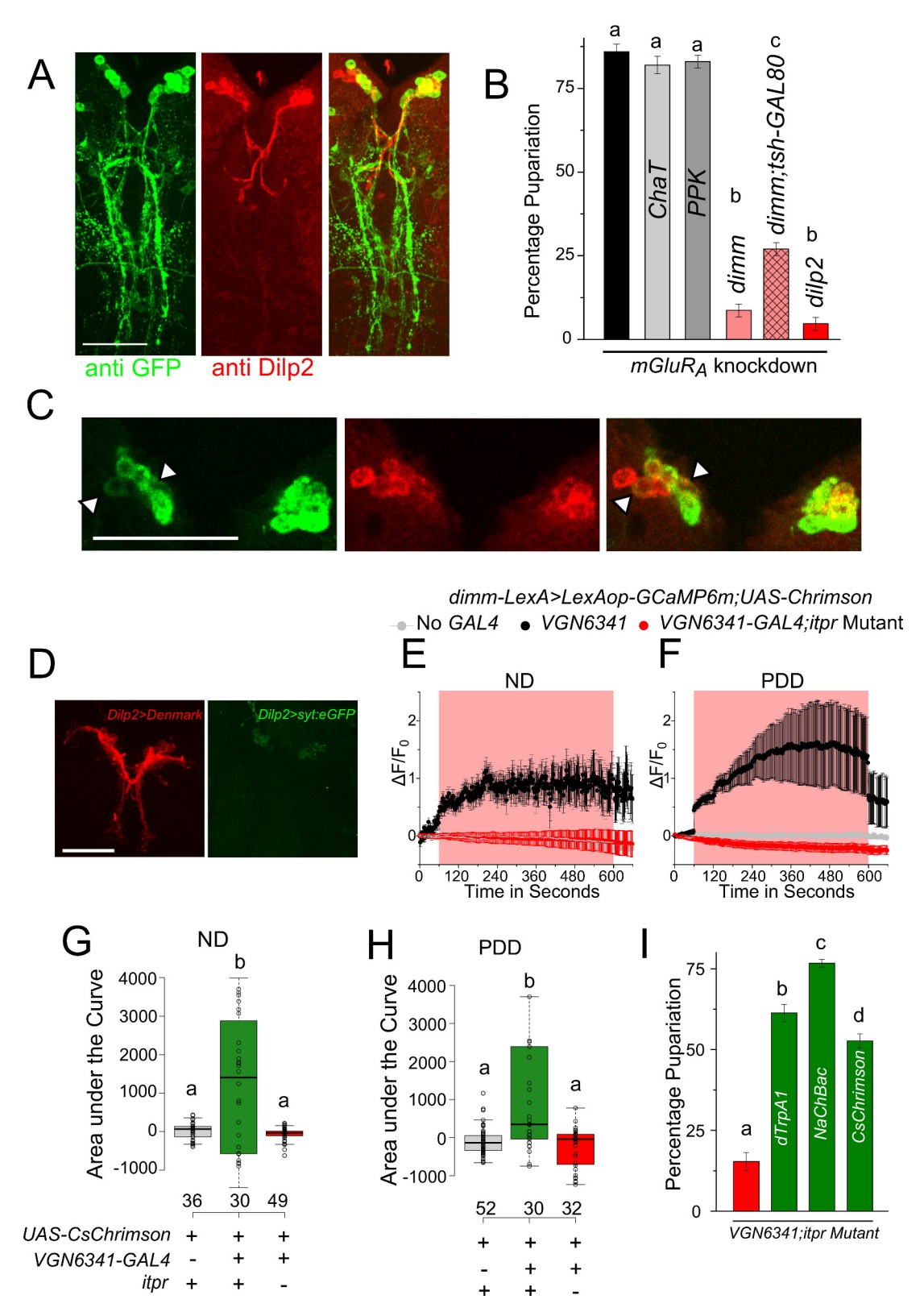

**Figure 5.** Glutamatergic neurons in the larval ventral ganglion convey signals to peptidergic neurons of the mNSC. (**A**) High-magnification images of the mNSC area in a GRASP experiment between the peptidergic and the glutamatergic domains stained for GFP and Dilp2. (**B**) Bars represent mean percentage pupariation (± SEM) of larvae subjected to *mGluR_A* knockdown using indicated *GAL4* drivers on PDD. N ≥ 6 batches with 25 larvae each.
*Figure 5 continued on next page*

*Figure 5 continued*

(**C**) A z-project of selected substacks at higher magnification showing the mNSC region from (**A**). Scale bar indicates 10 µm. Arrow heads point to weakly stained cells. (**D**) Confocal images showing the mNSC region of the *Dilp2-GAL4* simultaneously driving an axonal and dendritic marker (*Dilp2>UAS-DenMark, UAS-SyteGFP*). **E** and **F** Traces represent time series of mean normalized GCaMP responses (± SEM) from peptidergic cells in the mNSC of the indicated genotypes on ND (**E**) or PDD (**F**). **G** and **H** Quantification of area under the curve from (**E**) and (**F**). Box plots and symbols are as described for *Figure 3F*. (**I**) Bars represent mean percentage pupariation (± SEM) of indicated genotypes subjected to PDD. N ≥ 6 batches with 25 larvae each. Scale bars indicate 50µm unless specified otherwise Bars with the same alphabet represent statistically indistinguishable groups (one-way ANOVA with a post hoc Tukey's test p<0.05).

The following figure supplements are available for figure 5:

**Figure supplement 1.** IP$_3$R signaling in glutamatergic neurons is required for activation of medial neurosecretory cells.

**Figure supplement 2.** Neurons of the mVG activate peptidergic neurons in the mNSCs.

neurons and not by direct knockdown of *mAChR* in peptidergic neurons with *dimm-GAL4* (*Figure 6G*). Thus, cholinergic stimulation of glutamatergic neurons in the mVG appears to regulate peptide release from the mNSCs by activating mGluR$_A$ (*Figure 6G*). Further support for the ability of glutamatergic neurons to stimulate peptidergic release from the mNSCs comes from optogenetic stimulation of *VGN6341*-marked glutamatergic neurons (*Figure 6D*). Enhanced release of ANF::GFP was observed during the period of optogenetic stimulation (*Figure 6H*). Conversely, acute optogenetic inhibition of *VGN6341*-marked glutamatergic neurons using halorhodopsin, a light-activated chloride pump known to hyperpolarize neurons (*Berni et al., 2012*; *Inada et al., 2011*), during CCh stimulation inhibited peptide release from mNSCs (*Figure 6E and I*).

Although somatic peptide release is known (*De-Miguel and Nicholls, 2015*; *Trueta and De-Miguel, 2012*), we also tested if peptide release was affected more specifically in varicosities of the mNSCs that project to the ring gland. In the *itpr* mutant, ANF::GFP release was significantly reduced upon stimulation with 50 µm carbachol (*Figure 7A,C and E*). The extent of release could be rescued by expression of *mAChR*$^+$ in glutamatergic neurons marked by *VGN6341* (*Figure 7D and E*). Similar results were observed when expression of ANF::GFP was restricted to a subset of mNSCs marked by *dilp2-GAL4* (*Figure 7F–H*).

### *Dilp2* release on PDD is regulated by glutamatergic neurons in the mVG

Among the peptides released from the mNSCs, Dilp2 has been implicated in the response to protein deficiency (*Géminard et al., 2009*; *Sano et al., 2015*). Therefore, we measured Dilp2 levels in the mNSCs of larvae subjected to either normal or protein-deficient diets before and after subjecting them to CCh stimulation for 30 min. Dilp2 levels were assessed by immunohistochemistry using a previously validated antibody (*Géminard et al., 2009*). Significant release of Dilp2 was observed post-CCh stimulation, as interpreted from the reduced staining observed in wild-type mNSCs of larvae on ND and PDD (*Figure 6A and B*). In the *itpr* mutant, however, CCh-stimulated Dilp2 release was significantly lower, irrespective of the diets

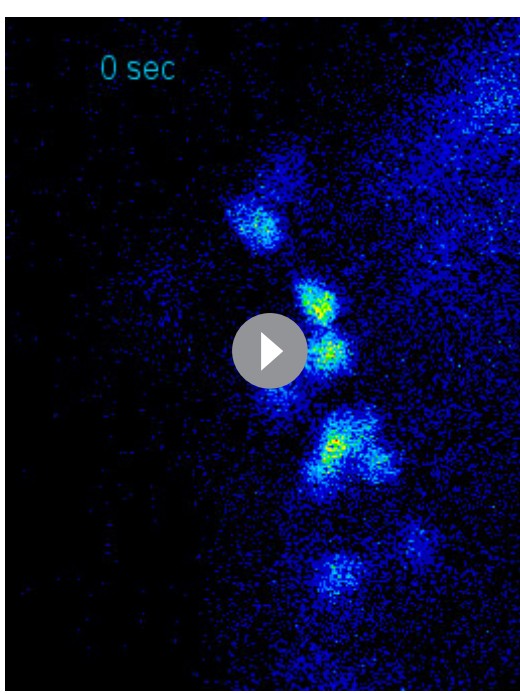

**Video 5.** Calcium transients observed in peptidergic neurons in the mNSCs of control larvae on PDD upon optogenetic activation of *VGN 6341* neurons.

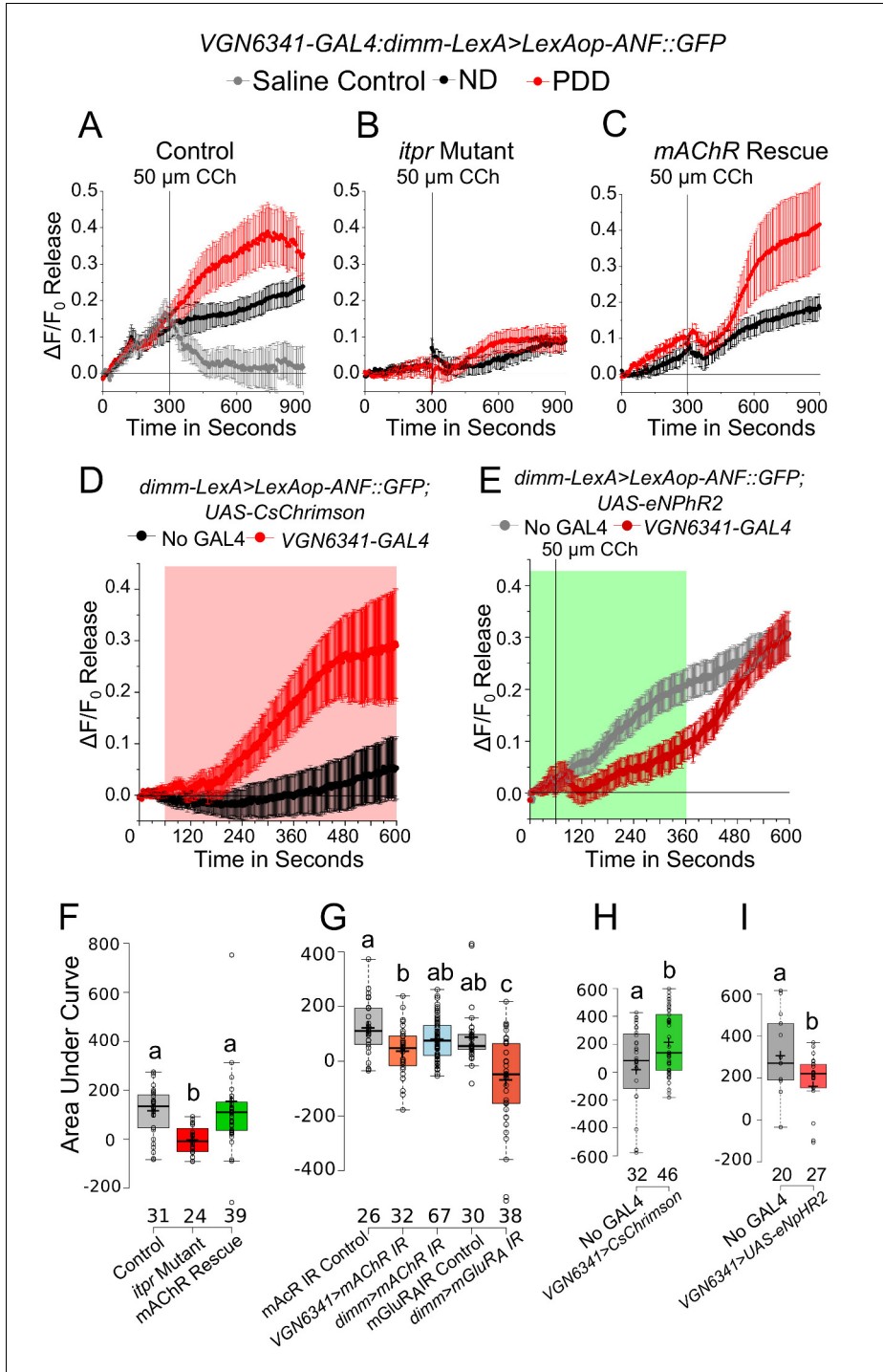

**Figure 6.** mAChR stimulation in glutamatergic neurons modulates enhanced peptide release from the mNSCs upon protein-deprivation. (A–C) Traces represent a time series of mean normalized peptide release (ANF::GFP; ± SEM) from mNSCs of the indicated genotypes upon Carbachol (CCh) stimulation. **D** and **E** Traces represent a time series of mean normalized peptide release (ANF::GFP; ± SEM) on PDD from the mNSCs upon optogenetic activation of *VGN6341-GAL4* (**E**) and using CCh under acute inhibition from the *VGN6341-GAL4* (**F**). Red and green boxes indicate duration of activation and inhibition, respectively. (**F–I**) Box plots of CCh stimulated peptide release (ANF::GFP) quantified by area under the curve from the mNSCs of the indicated genotypes on PDD. Box plots and symbols are as described for *Figure 3F*. Bars with the same alphabet represent statistically indistinguishable groups (one-way ANOVA with a post hoc Tukey's test p<0.05).

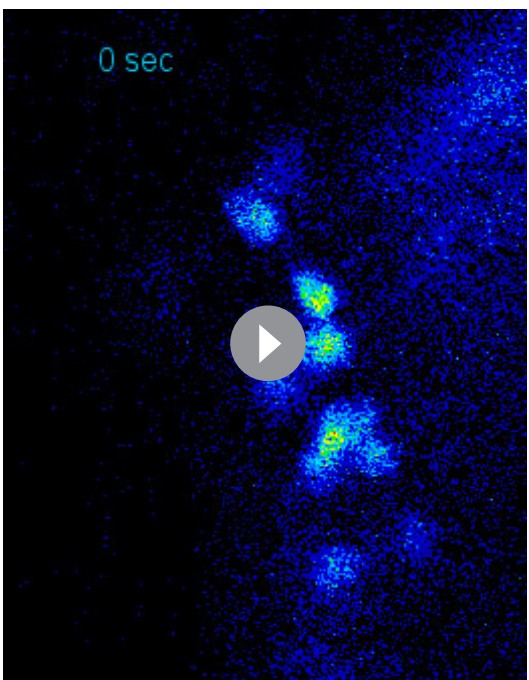

**Video 6.** Peptide release from mNSC of control larvae on PDD as observed by the decrease in ANF::GFP intensity upon addition of CCh. Green flash indicates point of addition of CCh.

(*Figure 8A and B*). CCh stimulation of brains with *itpr* knockdown in neurons marked by *VGN6341* also induced weaker release of Dilp2 (*Figure 8C*). Taking these observations forward, when *Dilp2*+-was over-expressed using *Dilp2-GAL4*, or when *Dilp2* positive neurons were maintained in an excitable state by expression of NaChBac, a partial rescue of pupariation in the *itpr* mutant was observed on PDD (*Figure 8D*). Taken together, these results suggest that pupariation on a protein-deficient diet requires mVG-mediated neuropeptide release from the mNSCs. Alternately, or in parallel, pupariation on PDD might require up-regulation of Dilp2 synthesis in the mNSCs, triggered by mGluR$_A$ signaling. However this seems unlikely because *Dilp2* mRNA levels are reduced to equal extents upon starvation in both wild-type and *itpr* mutant brains (*Figure 8—figure supplement 1*).

## IP$_3$R signaling in glutamatergic neurons of the mVG regulate transcription of ecdysone biosynthetic genes in the ring gland during protein-deprivation

A peak of ecdysone release late in the wandering third-instar triggers larval pupariation on ND (*Warren et al., 2006*). Expression levels of genes encoding enzymes of the ecdysteroid biosynthetic pathway have been studied (*McBrayer et al., 2007*; *Shimada-Niwa and Niwa, 2014*), in larvae on a normal diet. Specifically, transcripts of *shadow (sad), spookier (spok), phantom (phm), neverland (nvd)* and *disembodied (dib)* are up-regulated dramatically before pupariation (*McBrayer et al., 2007*; *Shimada-Niwa and Niwa, 2014*; *Warren et al., 2006*). It is therefore likely that the neural circuit identified for pupariation on PDD affects ecdysone synthesis by regulating ecdysteroid biosynthesis. Levels of *sad, spok, phm, nvd* and *dib* transcripts over time were characterized by analysis of RNA isolated from wild-type prothoracic glands from larvae on PDD. Transcripts for all these genes peaked approximately 42 hr post protein-deprivation in wild-type larvae aged 80–88 hr AEL (*Figure 9A*). The expression levels of these genes were lower in the *itpr* mutant at 42 hr and 66 hr and could be significantly rescued by overexpression of *mAChR*+in the neurons marked by *VGN6341*, especially at 66 hr (*Figure 9B*). Their up-regulation corresponded to a rescue in pupariation as well (*Figure 3A*). Taken together these data indicate that glutamatergic neurons of the mVG regulate expression of genes required for the ecdysone peak for pupariation on PDD in a *mAChR-* and *itpr*-dependent manner (*Figure 9C and D*). Ecdysteroid synthesis and release is regulated by neuropeptides, predominantly prothoracicotropic hormone (PTTH) (*McBrayer et al., 2007*; *Mirth et al., 2005*), as well as through insulin signaling (*Colombani et al., 2005*; *Mirth et al., 2005*). We propose that this regulation of ecdysone synthesis by the mVG is through neuropeptide release from the mNSCs.

## Discussion

In this study, we demonstrate that protein-deprivation signals are sensed by Pickpocket-expressing sensory neurons and conveyed to glutamatergic interneurons of the mid-ventral ganglion (mVG). These mVG neurons require mAChR-stimulated intracellular Ca$^{2+}$ release through the IP$_3$R to signal to downstream peptidergic neurons, including the medial neurosecretory cells. On a protein-deficient diet, these connections stimulate release of neuropeptides, which in turn, regulate expression of genes required for ecdysteroid biosynthesis to allow pupariation (*Figure 9D*). Calcium responses

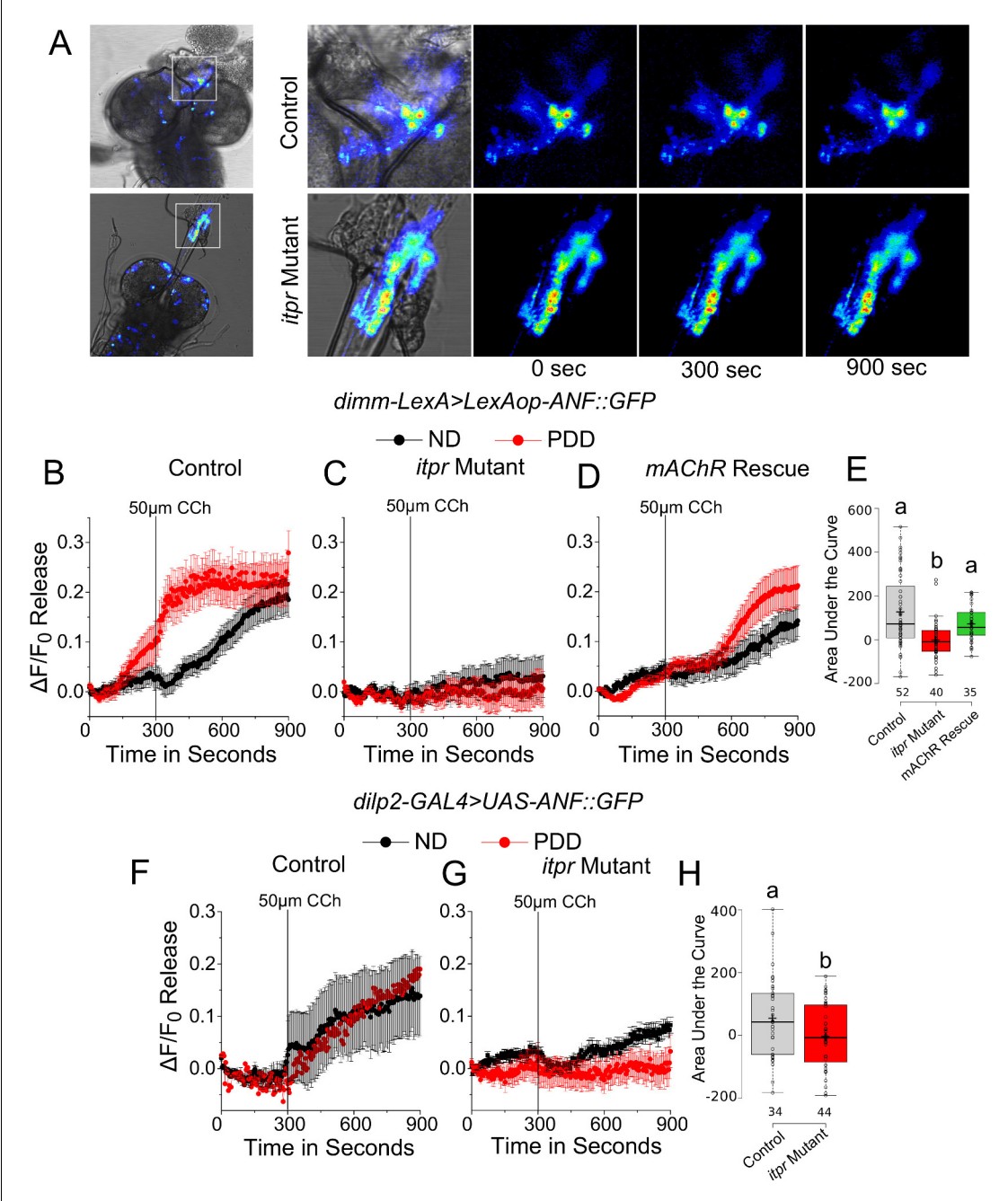

**Figure 7.** mAChR stimulation of glutamatergic neurons modulates peptide release from varicosities at the ring gland upon protein-deprivation. (**A**) Time series of ANF::GFP release from varicosities in the ring gland of the indicated genotypes at the indicated time intervals, after stimulation by Carbachol (CCh). (**B–D**) Traces represent a time series of mean normalized peptide ( ± SEM) release from varicosities at the ring glands of the indicated genotypes after Carbachol (CCh) stimulation. (**E**) Box plots representing CCh-stimulated peptide release with ANF::GFP quantified by area under the curve of the indicated genotypes on PDD from (**B–D**). Box plots and symbols are as described for *Figure 3F*. **F** and **G** Traces represent a time series of mean normalized peptide release (ANF::GFP; ± SEM) from varicosities at the ring glands of indicated genotypes upon Carbachol (CCh) stimulation. (**H**) Box plots representing CCh-stimulated peptide release with ANF::GFP quantified by area under the curve of the indicated genotypes on PDD from (**F** and **G**.). Box plots and symbols are as described for *Figure 3F*. Bars with the same alphabet represent statistically indistinguishable groups (one-way ANOVA with a post hoc Tukey's test p<0.05).

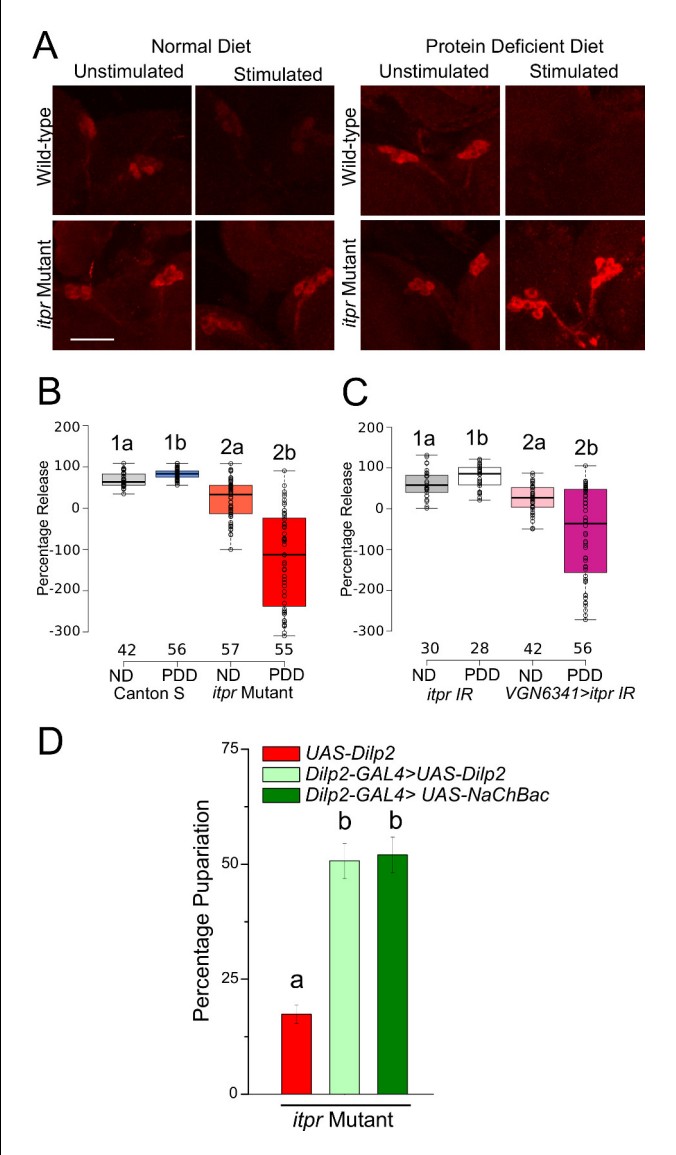

**Figure 8.** Glutamatergic neurons regulate Dilp2 release upon protein-starvation. (**A**) Dilp2 staining in larval brains from the indicated genotypes before and after stimulation with 50μM CCh for 30 min. (**B**) and (**C**) Box plots representing percentage release of Dilp2 from the respective genotypes subjected to ND or PDD. Box plots and symbols are as described for *Figure 2F*. A significant interaction was observed between genotype and diet (p<0.001). (**D**) Bars represent percentage pupariation as mean ± SEM of indicated genotypes on PDD. N ≥ 6 batches with 25 larvae each. Bars with the same alphabet represent statistically indistinguishable groups (one-way ANOVA with a post hoc Tukey's test p<0.05). For two-way ANOVA, numbers represent the variable genotype and alphabets represent diets (p<0.05).

The following figure supplement is available for figure 8:

**Figure supplement 1.** *dilp2* mRNA levels are not altered in the *itpr* mutant.

of mVG glutamatergic neurons differ between the normal and protein-deficient diets, suggesting that nutrient-dependent changes take place in these neurons that affect their cellular signaling properties. Knockdown of either *mAChR* or *itpr* in the mVG neurons blocks the functional connectivity between these neurons and the mNSCs (*Figure 9C*). The loss of this connectivity affects neuropeptide release from the mNSCs that is required for pupariation under protein-deprivation.

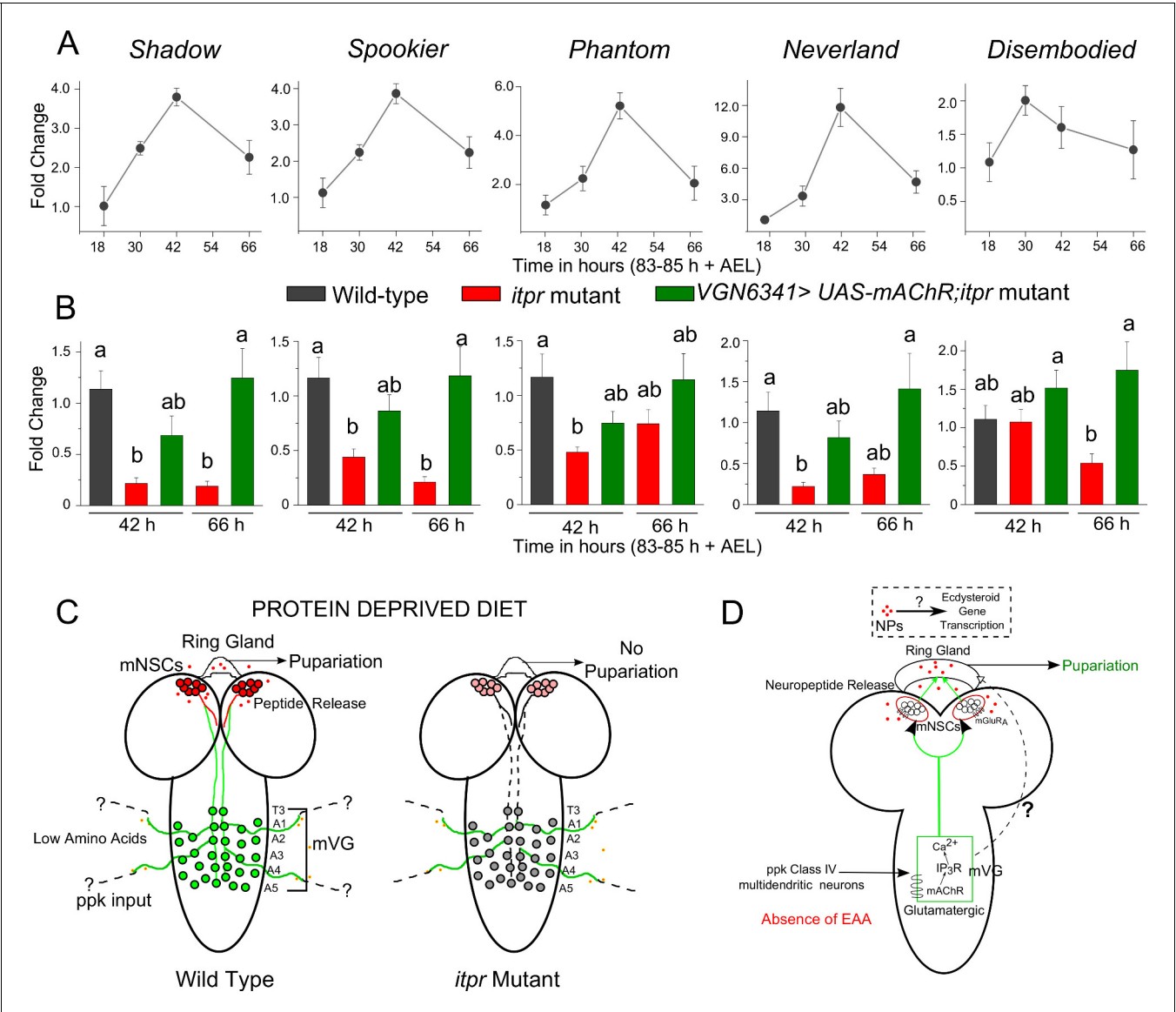

**Figure 9.** IP$_3$R signaling in glutamatergic neurons regulates the expression of ecdysone biosynthetic genes during protein-deprivation. (**A**) Normalized fold changes in the mRNA levels of the indicated genes represented as means ± SEM at indicated time points after 83–85 hr AEL on PDD from wild-type ring glands (n ≥ 3). (**B**) Bars represent mean fold changes (± SEM) of expression levels of respective ecdysteroid-synthesizing genes as shown in (**A**) from the ring glands of indicated genotypes at indicated time points (n ≥ 5). Bars with the same alphabet represent statistically indistinguishable groups (one-way ANOVA with a post hoc Tukey's test p<0.05). (**C**) Schematics of the neuronal circuit required for pupariation under protein-deprivation in early third instar larvae (80–88 hr AEL). Upon amino acid deprivation, glutamatergic neurons of the mVG are activated by ppk inputs. These glutamatergic neurons activate peptidergic cells in the mNSC to release peptides to further modulate ecdysteroid gene expression. In *itpr* mutants upon amino acid deprivation, glutamatergic inputs from the mVG to the mNSCs remain silent. (**D**) Schematic model of the signaling mechanisms observed in the circuit for pupariation under protein-deprived conditions.

Glutamatergic signals from the mVG are apparently not essential for pupariation on a normal diet, indicating that the likely function of this circuit is an adaptation to uncertain availability of food in nature.

## Neuronal circuit for nutrient sensing

Similar to other insect larvae, the larval phase in *Drosophila* is dedicated to feeding and attaining a nutritional state ready for metamorphosis (*Nijhout, 2003*). Feeding and nutrient sensing thus

constitute important aspects of larval life. The glutamatergic interneurons identified here exhibit diet-induced plasticity in their response to carbachol, an acetylcholine mimic. Their response appears integral to circuit function, and very likely drives enhanced neuropeptide secretion from the mNSCs and modulates ecdysteroid synthesis in the ring gland. Unlike the central neurons described recently (*Bjordal et al., 2014*), the glutamatergic interneurons do not sense amino acid levels directly. Instead, they reside in the ventral ganglion and receive inputs from sensory neurons of the class-IV multidendritic type marked by the *ppk-GAL4*. The precise identity of ppk-positive cells that sense the lack of amino acids needs to be determined.

Vertebrate neurosecretory cells located in the hypothalamus are integral to nutrient-sensing and energy homeostasis (*Sternson, 2013*) and have been equated to the mNSCs (*Hartenstein, 2006*). A recent report identified the importance of peripheral inputs to the central brain for maintaining nutritional homeostasis (*Zeng et al., 2015*). There is thus a remarkable similarity in the nutrient-sensing circuit we describe here to the circuits proposed in mammals. In vertebrates, neuronal circuit perturbation involving peripheral and central circuits remains challenging. The glutamatergic neurons identified here appear to regulate neuropeptide release from the mNSCs and might be equivalent to neuronal regulators of hypothalamic neurons.

## Role of IP$_3$R in metabolic adaptation

Intracellular Ca$^{2+}$ release through the IP$_3$R has been implicated in the regulation of lipid and carbohydrate metabolism in vertebrates and invertebrates (*Agrawal et al., 2009*; *Ozcan et al., 2012*; *Wang et al., 2012*). Vertebrate IP$_3$R is encoded by three genes (*Furuichi et al., 1989*; *Yamamoto-Hino et al., 1994*), thus creating the possibility for layered, complex regulation. Mice null for IP$_3$R2 and IP$_3$R3 exhibit digestive defects associated with loss of exocrine secretion from pancreatic β cells and the salivary gland (*Futatsugi et al., 2005*). Intracellular calcium signaling in the context of neuronal regulation of a systemic response to nutritional cues as described here, remains to be tested in vertebrates. We speculate that IP$_3$R-mediated calcium release in glutamatergic interneurons stimulates glutamate release onto presynaptic terminals of neurosecretory cells. We propose that reducing IP$_3$R function in glutamatergic neurons in the mVG alters their connectivity with peptidergic neurons. Synapse formation induced by glutamate release has been reported (*Kwon and Sabatini, 2011*). Our data do not have the resolution to distinguish between physical and molecular bases of synaptic dysfunction. Activity-dependent rescue (observed in *Figure 4I*) restores the connectivity (*Figure 5—figure supplement 1H*), suggesting that these molecular changes, at least in part affect excitability.

In addition to mAChR, our screen identified eleven other GPCRs that signal through intracellular Ca$^{2+}$ and whose knockdown in glutamatergic neurons resulted in loss of pupariation on PDD. Although each GPCR leads to the same systemic phenotype, the mechanisms in each case might differ, and each requires further study. Neuropeptide receptors constitute 8 out of 12 of these receptors. Interestingly, the neuropeptide receptors we report here, except SIFaR, have all been implicated in regulating feeding in vertebrates and insects (*Caers et al., 2012*; *Hentze et al., 2015*; *Sano et al., 2015*).

The calcium transients observed in glutamatergic neurons in the mVG upon optogenetic activation of cholinergic inputs were markedly different in larvae on ND and PDD. On PDD, the transients were oscillating, whereas on ND oscillations were either dampened or absent. Moreover, oscillations in the glutamatergic neurons appear to drive oscillations in the mNSCs of larvae on PDD as seen by the increase in neurons that oscillate (*Figure 5—figure supplement 1I*). Such oscillations are known to cause exocytosis of neuropeptides (*Tse et al., 1993*) and may underlie enhanced neuropeptide secretion required for pupariation on PDD. The molecular basis of such diet-induced plasticity leading to robust oscillatory activity in the glutamatergic neurons of the mVG needs investigation.

In a recent report, dopaminergic neurons in the larval brain of *Drosophila* were demonstrated as directly sensing amino acids leading to their activation and consequent changes in food-intake behaviour on an amino acid deficient medium (*Bjordal et al., 2014*). These findings, however, do not explain the full range of responses to amino acid deprivation. Our results describe a response mechanism that organisms employ to overcome the developmental consequences of protein-deprivation (*Chen et al., 2015*).

## Context-dependent peptide and hormonal control allows pupariation

Higher levels of Dilp2 were observed in the larval mNSCs of IP$_3$R mutants, as well as in larvae with IP$_3$R knockdown in glutamatergic neurons of the ventral ganglion. We attribute the excess Dilp2 to reduced release of Dilp2 from the mNSCs. The absence of up-regulation of *dilp2* transcripts in IP$_3$R mutants supports this idea (*Figure 8—figure supplement 1*). This is, to our knowledge, the first report of Dilp2 regulation by neurons in the ventral ganglion. Environmental nutrients regulating Dilp function has been well documented (*Géminard et al., 2009*; *Kim and Neufeld, 2015*; *Rajan and Perrimon, 2012*). Most of these studies report remote sensing attributed to the fat body (*Géminard et al., 2009*; *Koyama and Mirth, 2016*; *Rajan and Perrimon, 2012*; *Sano et al., 2015*). The regulation we report here is another layer in the overall regulation of Dilps in *Drosophila* that seems to be particularly important upon protein-deprivation. Such complex layers of modulation are not surprising given that insulin signaling is important in different aspects of development and growth in *Drosophila* (*Colombani et al., 2005*). The Dilps have been speculated as regulators of pupariation (*Koyama et al., 2014*), and Dilp neurons project to the ring gland where ecdysone production occurs (*Cao and Brown, 2001*). Further work will be required to understand the regulation of Dilp secretion as a balance between neuronal and fat body signaling. Ultimately, all these layers of regulation seem to depend on the environmental context and developmental stage.

In insects, steroid hormones control developmental transitions (*Thummel et al., 1990*), including larval moults and metamorphosis. In third-instar larvae, there are ecdysone peaks ranging from several small ones to a bigger commitment peak prior to pupariation (*Warren et al., 2006*). Ecdysone is the switch for developmental change, hence needs to be tightly regulated through transcriptional control of ecdysteroid genes (*McBrayer et al., 2007*) and ecdysteroid biosynthesis is reported to be influenced by environmental conditions (*Shimada-Niwa and Niwa, 2014*). Serotonergic regulation of ecdysone release by changing neurite projections in a nutrient-dependent manner has been reported recently (*Shimada-Niwa and Niwa, 2014*). Our results do not rule out connectivity between mVG neurons and the supra-oesophageal ganglion in the central brain that could additionally stimulate the ring gland. We propose that the circuit identified here functions as a further layer of regulation, required during either sudden starvation or nutrient deprivation. It is required to make the key developmental decision of whether and when to pupariate. Energy homeostasis at a systemic level involves integrating environmental cues with internal states. The circuit we describe is such an integrator.

# Materials and methods

## Fly stocks and rearing

*Drosophila* strains were grown on cornmeal medium supplemented with yeast (ND) at 25°C unless otherwise noted. The protein-deprived diet (PDD) contained 100 mM sucrose with 1% agar. For optogenetic experiments, egg laying was carried out in cornmeal medium supplemented with 200 µM all-*trans*-retinal (ATR), and larvae were transferred at 84 ± 4 hr onto ND or PDD with 400 µM ATR. *Canton S* was used as wild-type (WT) throughout. A table of all stocks used is appended as *Supplementary file 1*. The *itpr IR* was used with *UAS-dicer* in all experiments. For the GPCR RNAi screen, RNAi lines were obtained from either VDRC or NIG fly stock centres. Larvae at 84 ± 4 hr post egg laying were transferred to PDD or ND in batches of 25 and were scored for pupariation. At least six independent batches were performed for each genotype on each media. These are reported as percentage pupariation. For experiments involving diet-based rescues, PDD was supplemented either EAA (1x MEM EAA, GIBCO) or growth supplements (5x RPMI 1640 Amino acid solution, Sigma). For rate of pupariation, all genotypes in *Figure 1* were monitored every 12 hr after transfer.

## Immunohistochemistry

Immunostaining of larval *Drosophila* brains was performed as described previously (*Daniels et al., 2008*). Briefly, larval brains were dissected in 1x phosphate buffered saline (PBS) and fixed with 4% Paraformaldehyde or Bouin's fixative for dvGlut staining. They were washed three to four times with 0.2% phosphate buffer, pH 7.2 containing 0.2% Triton-X 100 (PTX) and blocked with 0.2% PTX containing 5% normal goat serum (NGS) for four hours at 4°C. Respective primary antibodies were incubated overnight (14–16 hr) at 4°C. For dvGlut staining, the brains were incubated for 60–72 hr at

4°C. After washing three to four times with 0.2% PTX at room temperature, they were incubated in the respective secondary antibodies for 2 hr at room temperature. The following primary antibodies were used: rabbit anti-GFP antibody (1:10,000; A6455, Life Technologies, RRID:AB_221570), mouse anti-GFP antibody (1:50; Santa Cruz Biotechnology, RRID:AB_627695), rabbit anti-dsRed (1:500; 632496, Clontech, RRID:AB_10015246) mouse anti-nc82 (anti-brp) antibody (1:50; a kind gift from Erich Buchner, RRID:AB_2314869,), rabbit anti-dvGlut (1:1000; a kind gift from Aaron DiAntonio, RRID:AB_2314346), rat anti-dilp2 (1:400; a kind gift from Pierre Leopold). Secondary antibodies were used at a dilution of 1:400 as follows: anti-rabbit Alexa Fluor 488 (#A11008, Life Technologies, RRID:AB_143165), anti-mouse Alexa Fluor 488 (#A11001, Life Technologies, RRID:AB_141367), anti-mouse Alexa Fluor 568 (#A11004, Life Technologies, RRID:AB_141371), anti-rabbit Alexa Fluor 594 (#A11037, Life Technologies, RRID:AB_10561549) and anti-rat Alexa Fluor 633 (#A21094, Life Technologies, RRID:AB_10561523). Confocal images were obtained on the Olympus Confocal FV1000 microscope (Olympus) with a 40x, 1.3 NA objective or with a 60x, 1.4 NA objective. Images were visualized using either the FV10-ASW 4.0 viewer (Olympus) or Fiji (RRID:SCR_002285) (*Schindelin et al., 2012*).

## Live imaging from larval brains

Larval brains were dissected in hemolymph-like saline (HL$_3$) (70 mM NaCl, 5 mM KCl, 20 mM MgCl$_2$, 10 mM NaHCO$_3$, 5 mM trehalose, 115 mM sucrose, 5 mM HEPES, 1.5 mM Ca$^{2+}$, pH 7.2), embedded in 0.2% low-melt agarose (Invitrogen), and bathed in HL3. GCaMP6m was used as the genetically encoded calcium sensor. ANF::GFP was expressed genetically to quantify vesicular release. Images were taken as a time series on an XY plane at an interval of 4 s using a 20x objective with an NA of 0.7 on an Olympus FV1000 inverted confocal microscope (Olympus Corp., Japan). For thermogenetic experiments, a heated stage was used to shift the temperature to 30°C to activate TrpA1. For optogenetic stimulation, a 633-nm laser line was used for activation of CsChrimson while simultaneously acquiring images with the 488 nm laser line, and the images were acquired every 1.5 s. For channelrhodopsin activation 488 nm laser line was used. For optogenetic inhibition experiments a green 543-nm laser line was driven simultaneously with image acquisition using the 488 nm laser line. All live imaging experiments were performed with at least five independent brain preparations and the exact number of cells for each experiment are indicated in the figures.

The raw images were extracted using Image J1.48 and regions of interest (ROI) selected using the Time Series Analyser plugin. $\Delta F/F$ was calculated using the formula $\Delta F/F = (F_t - F_0)/F_0$, where $F_t$ is the fluorescence at time t and $F_0$ is baseline fluorescence corresponding to the average fluorescence over the first ten frames. Area under the curve was calculated from the point of stimulation which was considered as 0$^{th}$ second for stimulation up to 300 s using Microsoft Excel (Microsoft) and plotted using BoxPlotR (*Spitzer et al., 2014*). $\Delta F/F$ Release was calculated as $(F_0 - F_t)/F_0$ where $F_t$ is the fluorescence at time t and $F_0$ is baseline fluorescence corresponding to the average fluorescence over the first 10 frames. Area under the curve was calculated from 0s to 600s using Microsoft Excel (Microsoft), and box plots were plotted using BoxPlotR (*Spitzer et al., 2014*). For experiments with UAS-Shi$^{ts}$, a heated microscopic stage was used.

## Quantification of Dilp2 release

Larval brains on ND or PDD were dissected and stained for Dilp2. Image acquisition was performed using similar acquisition settings and were processed using Fiji. Cells and background were marked using an ROI selection plugin and then the intensity across the stacks was measured. Brightest values were obtained using a Max function in Microsoft Excel. Intensity values were calculated for each cell in brains irrespective of ND or PDD, by subtracting background. To then obtain percentage release upon stimulation, intensity of cells were normalized to average intensity on the same diet. At least five independent brain preparations per genotype per condition were used and the exact number of cells are indicated in the figure.

## RNA isolation and quantitative PCR

Flies were transferred every 2 hr to obtain larvae that were very tightly staged. Ring glands from larvae of the appropriate genotype and age were dissected in phosphate buffer saline prepared in double distilled water treated with diethyl pyrocarbonate (Sigma). Each sample consisted of five

Ring glands or 5 CNS and these were homogenized in 500 µl TRIzol per sample by vortexing immediately after dissection. At least three biological replicate samples were made for each genotype. After homogenization the sample was kept on ice and processed within 30 min or stored at −80°C until processing for up to 4 weeks. RNA was isolated by following manufacturer's protocol for TRIzol (Ambion, ThermoFischer Scientific). Purity of the isolated RNA was estimated by NanoDrop spectrophotometer (Thermo Scientific) and integrity was checked by running it on a 1% Tris-EDTA agarose gel.

Approximately 100 ng of total RNA was used per sample for cDNA synthesis. DNAse treatment and first strand synthesis were performed as described previously (*Pathak et al., 2015*). Quantitative real time PCRs (qPCRs) were performed in a total volume of 10 µl with Kapa SYBR Fast qPCR kit (KAPA Biosystems, Wilmington, MA) on an ABI 7500 fast machine operated with ABI 7500 software (Applied Biosystems). Technical duplicates were performed for each qPCR reaction. A melt analysis was performed at the end of the reaction to ensure the specificity of the product. The fold change of gene expression in any experimental condition relative to wild-type was calculated as $2^{-\triangle\triangle Ct}$ where $\triangle\triangle Ct$ = (Ct (target gene) –Ct (rp49)) $_{Expt.}$ - (Ct (target gene) – Ct (rp49)) $_{Control.}$

*rp49* was used as the internal control and the primer sequences used are provided in *Supplementary file 2*. Primers for genes of the ecdysteroid biosynthetic pathway have been described previously (*Shimada-Niwa and Niwa, 2014*).

## Generation of transgenic flies

The *dimmed-LexA::p65* construct was created using recombineering techniques based on those of (*Warming et al., 2005*). 5′ and 3′ homology arms comprising about 200 bases each of NLS::LexA and the HSP70 terminator were amplified from *pBPnlsLexA::p65Uw*, a gift of Gerald Rubin (*Pfeiffer et al., 2010*) and inserted into *pSK+-rpsL-kana* (*Wang et al., 2009*) to create a selectable generic landing cassette. Primers carrying gene-specific homology arms to target the cassette to the first coding exon of *dimmed* were used to PCR-amplify this cassette, and the resulting *dimmed*-flanked marker cassette was recombined into P[acman] BAC clone CH321-46B06 (*Venken et al., 2009*) (obtained from Children's Hospital Oakland Research Institute, Oakland, CA), replacing the coding portion of the first *dimmed* coding exon while leaving intact the endogenous 5′ UTR as well as the following introns and exons (although these are presumably no longer transcribed because of the inserted terminator sequences). The landing-site cassette was then replaced via a second recombination with full-length *LexA::p65-HSP70*, also amplified from *pBPnlsLexA::p65Uw*. The recombined regions of the BAC were sequence-verified, and the finished BAC was integrated into *attP* site *VK00033* (*Venken et al., 2006*) on chromosome arm 3L by Genetic Services, Inc. (Cambridge, MA).

For the *VGN6341-LexA*, the 505 bp fragment of the *dvGlut* gene enhancer as in *VGN6341-GAL4* (*Syed et al., 2016*) was PCR amplified from wild-type DNA, the sequence was verified and cloned into *pDONR-221-p1-p5r* (Invitrogen) to get an entry clone by performing the BP reaction. This entry clone along with the *LexA* entry clone and the destination vector were combined in an LR reaction to generate *VGN6341-LexA*.

For the LexAop-ANF::GFP clone, the sequence of ANF:GFP was PCR amplified from DNA isolated from UAS-ANF::GFP flies and cloned into pDONR-221-p5-p2 (Invitrogen) to get an entry clone by performing the BP reaction. This entry clone along with the LexAop entry clone and the destination vector were combined in an LR reaction to generate LexAop-ANF::GFP.

The entry clones *pENTR L5-LexAp65-L2* (41437) and *pENTR L1-13XLexAop2-R5* (41433) and the destination vector *pDESTsvaw* (32318) were obtained from Addgene. The BP and LR reactions were performed using the Multisite Gateway Pro cloning kit (Invitrogen, 12537–102) following the half volume protocol described in (*Petersen and Stowers, 2011*). All primer sequences are listed in *Supplementary file 2*.

## Statistics

All statistical tests are mentioned in the figure legends and were performed using Origin 8.0. *Supplementary file 4* has all statistical tests and their p-values.

## Acknowledgements

We thank, Yoshi Aso, Claude Desplan, Amita Sehgal, Pierre Leopold, Aaron DiAntonio for fly stocks and antibodies; the Vienna Drosophila RNAi Center (VDRC), National Institute of Genetics Fly Stock Center and the Bloomington Stock Center (National Institutes of Health P40OD018537) for flies. We thank Steve Stowers for plasmids and Matthias Landgraf for introducing us to them. We also thank the Fly Facility, the Sequencing Facility and the Central Imaging and Flow Cytometry Facility at NCBS.

## Additional information

### Funding

| Funder | Grant reference number | Author |
| --- | --- | --- |
| National Centre for Biological Sciences | Core funding | Gaiti Hasan |
| Council of Scientific and In-dustrial Research | Graduate fellowship | Siddharth Jayakumar |
| National Centre for Biological Sciences | Graduate fellowship | Shlesha Richhariya |

The funders had no role in study design, data collection and interpretation, or the decision to submit the work for publication.

### Author contributions

SJ, SR, Conception and design, Acquisition of data, Analysis and interpretation of data, Drafting or revising the article; OVR, Acquisition of data, Contributed unpublished essential data or reagents; MJT, Drafting or revising the article, Contributed unpublished essential data or reagents; GH, Conception and design, Analysis and interpretation of data, Drafting or revising the article, Contributed unpublished essential data or reagents

### Author ORCIDs

Michael J Texada, http://orcid.org/0000-0003-2479-1241
Gaiti Hasan, http://orcid.org/0000-0001-7194-383X

## Additional files

### Supplementary files

• Supplementary file 1. List of fly stocks.

• Supplementary file 2. List of primer sequences.

• Supplementary file 3. List of all GPCR RNAi lines tested.

• Supplementary file 4. Excel sheet with all statistical tests and their p-values.

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
