## [Decision Letter]

Thank you for submitting your article "*Drosophila* larval to pupal switch under nutrient stress requires IP3R/Ca^2+^ signalling in glutamatergic interneurons" for consideration by *eLife*. Your article has been reviewed by three peer reviewers, one of whom, Leslie Griffith is a member of our Board of Reviewing Editors, and the evaluation has been overseen by Eve Marder as the Senior Editor.

The reviewers have discussed the reviews with one another and the Reviewing Editor has drafted this decision to help you prepare a revised submission.

Summary:

This is a very interesting and complete characterization of the circuit that allows animals to pupate in the face of amino acid deprivation. Jayakumar and colleagues describe a neuronal circuit linking class IV multidendritic neurons to peptidergic median neurosecretory neurons (mNSCs) via glutamatergic neurons. Importantly, they show that the functioning of this circuit depends on IP3R-dependent calcium signalling, and present a link to ecdysteroid signalling that is timing pupariation. The finding and characterisation of a neuronal circuit designated to overcome nutrient stress to enable developmental transitions is certainly a major achievement, and will be of interest to many.

Essential revisions:

The overall study seems sound, but there are a number of places where the interpretations or data presentation are not clear and the paper should be improved.

1) In particular, the anatomical evidence for direct connections is weak in a couple places. Specific concerns and questions about interpretation and clarity can be found in the excerpts of the individual reviews, which are appended. The most critical, and hopefully not difficult to obtain, additional data that would help clarify the circuit would be to:

A) Try to establish direct connectivity between the mVG cells in *VGN6341-GAL4* and mNSCs using immunostaining.

B) Examine ANF-GFP release at actual varicosities as opposed to the soma.

2) The current manuscript glosses over some of the areas of confusion. In particular, the model at the end is almost completely uninformative. It would be helpful to have an explicit hypothesis of the circuitry and signals being transmitted and to pinpoint which elements of this model they have been able to confirm experimentally and which remain unclear. In short, the rich detail of this study needs to be critically linked to an explicit working model. Holes that remain should be discussed candidly.

---

## [Author Response]

*The overall study seems sound, but there are a number of places where the interpretations or data presentation are not clear and the paper should be improved.*

*1) In particular, the anatomical evidence for direct connections is weak in a couple places. Specific concerns and questions about interpretation and clarity can be found in the excerpts of the individual reviews, which are appended. The most critical, and hopefully not difficult to obtain, additional data that would help clarify the circuit would be to:*

*A) Try to establish direct connectivity between the mVG cells in VGN6341-GAL4 and mNSCs using immunostaining*

We now provide anatomical evidence for connectivity between the two domains in Figure 4 and Video 4. The two domains (*VGN6341-GAL4* and *dimm-LexA*) expressing different fluorophores (GFP and *mCherry*) have been imaged at high resolution. In addition, *VGN6341-GAL4* driven GFP has been imaged in the background of *dilp2* promoter driven *mCherry*. In both cases we observe neurites emanating from the mVG and extending anteriorly to reach posterior projections of the mNSCs. The origin of anterior projecting neurites from *VGN6341* marked GFP neurons is difficult to distinguish in the z-project of the confocal image, because of multiple cell bodies and lateral projections from the peripheral mVG neurons to the midline of the ventral ganglion. However we traced the anterior projections stack by stack and these are shown in Video 4.

Independently we addressed this issue by activating the Restricted *VGN6341-GAL4* to test whether this activates the mNSCs. We do not observe mNSC activation in the absence of mVG cells activation. Thus mNSC activation is from *VGN6341-GAL4* marked glutamatergic neurons of the ventral ganglion and not the central brain. We have included these data in Figure 5—figure supplement 2.

*B) Examine ANF-GFP release at actual varicosities as opposed to the soma.*

ANF::GFP release has been measured at the varicosity sites at the ring gland. We have done this with varicosities marked by the *dimm-LexA* as well as the *dilp2-GAL4* drivers. These data are included additionally as a new Figure 7.

*2) The current manuscript glosses over some of the areas of confusion. In particular, the model at the end is almost completely uninformative. It would be helpful to have an explicit hypothesis of the circuitry and signals being transmitted and to pinpoint which elements of this model they have been able to confirm experimentally and which remain unclear. In short, the rich detail of this study needs to be critically linked to an explicit working model. Holes that remain should be discussed candidly.*

We have added two panels to the previously existing model showing the status of activity in the mVG glutamatergic neurons and the identified neural circuit in wild-type and *itpr* mutant conditions (Figure 9). The original panel has been retained to show the cellular signalling mechanism underlying the loss of circuit function (Figure 9). We have also identified and discussed the potential holes that remain in the Results and Discussion sections.